# PRE-TRAINING LiDAR-BASED 3D OBJECT DETECTORS THROUGH COLORIZATION

**Tai-Yu Pan**[1], **Chenyang Ma**[1], **Tianle Chen**[1], **Cheng Perng Phoo**[2], **Katie Z Luo**[2], **Yurong You**[2],
**Mark Campbell**[2], **Kilian Q. Weinberger**[2], **Bharath Hariharan**[2], and **Wei-Lun Chao**[1]
[1]The Ohio State University, Columbus, OH      [2]Cornell University, Ithaca, NY

## ABSTRACT

Accurate 3D object detection and understanding for self-driving cars heavily relies on LiDAR point clouds, necessitating large amounts of labeled data to train. In this work, we introduce an innovative pre-training approach, Grounded Point Colorization (GPC), to bridge the gap between data and labels by teaching the model to colorize LiDAR point clouds, equipping it with valuable semantic cues. To tackle challenges arising from color variations and selection bias, we incorporate color as "context" by providing ground-truth colors as hints during colorization. Experimental results on the KITTI and Waymo datasets demonstrate GPC's remarkable effectiveness. Even with limited labeled data, GPC significantly improves fine-tuning performance; notably, on just $20\%$ of the KITTI dataset, GPC outperforms training from scratch with the entire dataset. In sum, we introduce a fresh perspective on pre-training for 3D object detection, aligning the objective with the model's intended role and ultimately advancing the accuracy and efficiency of 3D object detection for autonomous vehicles. Code available: https://github.com/tydpan/GPC/

## 1 INTRODUCTION

Detecting objects such as vehicles and pedestrians in 3D is crucial for self-driving cars to operate safely. Mainstream 3D object detectors (Shi et al., 2019; 2020b; Zhu et al., 2020; He et al., 2020a) take LiDAR point clouds as input, which provide precise 3D signals of the surrounding environment. However, training a detector needs a lot of labeled data. The expensive process of curating annotated data has motivated the community to investigate model pre-training using unlabeled data that can be collected easily. Most of the existing pre-training methods are built upon contrastive learning (Yin et al., 2022; Xie et al., 2020; Zhang et al., 2021; Huang et al., 2021; Liang et al., 2021), inspired by its success in 2D recognition (Chen et al., 2020a; He et al., 2020b). The key novelties, however, are often limited to how the positive and negative data pairs are constructed. This paper attempts to go beyond contrastive learning by providing a new perspective on pre-training 3D object detectors.

We rethink pre-training's role in how it could facilitate the downstream fine-tuning with labeled data. A labeled example of 3D object detection comprises a point cloud and a set of bounding boxes. While the format is typical, there is no explicit connection between the input data and labels. We argue that a pre-training approach should help *bridge* these two pieces of information to facilitate fine-tuning.

*How can we go from point clouds to bounding boxes without human supervision?* One intuitive way is to first segment the point cloud, followed by rules to read out the bounding box of each segment. The closer the segments match the objects of interest, the higher their qualities are. While a raw LiDAR point cloud may not provide clear semantic cues to segment objects, the color image collected along with it does. In color images, each object instance often possesses a coherent color and has a sharp contrast to the background. This information has been widely used in image segmentation (Li & Chen, 2015; Liu et al., 2011; Adams & Bischof, 1994) before machine learning-based methods take over. We hypothesize that by learning to predict the pixel color each LiDAR point is projected to, the pre-trained LiDAR-based model backbone will be equipped with the semantic cues that facilitate the subsequent fine-tuning for 3D object detection. To this end, we propose pre-training a LiDAR-based 3D object detector by learning to colorize LiDAR point clouds. Such an objective makes the pre-training procedure fairly straightforward, which we see as a key strength. Taking models that

Figure 1: **Illustration of the proposed Grounded Point Colorization (GPC)**. We pre-train the detector backbone by colorization (left to right), taking the hints on a seed set of points (middle) to overcome the inherent color variation. We show that the middle step is the key to learning the semantic cues of objects from colors.

use point-based backbones like PointNet++ (Qi et al., 2017b;a) as an example, pre-training becomes solving a point-wise regression problem (for predicting the RGB values).

That said, the ground-truth colors of a LiDAR point cloud contain inherent variations. For example, the same car model (hence the same point cloud) can have multiple color options; the same street scene (hence the same point cloud) can be pictured with different colors depending on the time and weather. When selection bias occurs in the collected data (*e.g.*, only a few instances of the variations are collected), the model may learn to capture the bias, degrading its transferability. When the data is well collected to cover all variations without selection bias, the model may learn to predict the average color, downgrading the semantic cues (*e.g.*, contrast). Such a dilemma seems to paint a grim picture of using colors to supervise pre-training. In fact, the second issue is reminiscent of the "noise" term in the well-known bias-variance decomposition (Hastie et al., 2009), which cannot be reduced by machine learning algorithms. The solution must come from a better way to use the data.

Based on this insight, we propose a fairly simple but effective refinement to how we leverage color. Instead of using color solely as the supervised signal, we use it as "context" to ground the colorization process on the inherent color variation. We realize this idea by providing ground-truth colors to a seed set of LiDAR points as "hints" for colorization. (Fig. 1 gives an illustration.) On the one hand, this contextual information reduces the color variations on the remaining points and, in turn, increases the mutual information between the remaining points and the ground-truth colors to supervise pre-training. On the other hand, when selection bias exists, this contextual information directly captures it and offers it as a "shortcut" to the colorization process. This could reduce the tendency of a model to (re-)learn this bias (Lu, 2022; Clark et al., 2019; Cadene et al., 2019), enabling it to focus on the semantic cues useful for 3D object detection. After all, what we hope the model learns is not the exact color of each point but which subset of points should possess the same color and be segmented together. By providing the ground-truth colors to a seed set of points during the colorization process, we concentrate the model backbone on discovering with which points each seed point should share its color, aligning the pre-training objective to its desired role.

We implement our idea, which we name Grounded Point Colorization (GPC), by introducing another point-based model (*e.g.*, PointNet++) on top of the detector backbone. This could be seen as the projection head commonly added in contrastive learning (Chen et al., 2020a); Fig. 2 provides an illustration. While the detector backbone takes only the LiDAR point cloud as input, the projection head takes both the output embeddings of the LiDAR points and the ground-truth colors of a seed set of points as input and is in charge of filling in the missing colors grounded on the seed points. This leaves the detector backbone with a more dedicated task — producing output embeddings with clear semantic cues about which subset of points should be colored (*i.e.*, segmented) together.

We conduct extensive experiments to validate GPC, using the KITTI (Geiger et al., 2012) and Waymo (Sun et al., 2020) datasets. We show that GPC can effectively boost the fine-tuning performance, especially on the labeled-data-scarce settings. For example, with GPC, fine-tuning on 20% KITTI is already better than training 100% from scratch. Fine-tuning using the whole dataset still brings a notable 1.2% gain on overall AP, outperforming existing baselines.

Our contributions are three-fold. First, we propose pre-training by colorizing the LiDAR points — a novel approach different from the commonly used contrastive learning methods. Second, we identify the intrinsic difficulty of using color as the supervised signal and propose a simple yet well-founded and effective treatment. Third, we conduct extensive empirical studies to validate our approach.

## 2 RELATED WORK

**LiDAR-based 3D object detectors.** Existing methods can be roughly categorized into two groups based on their feature extractors. Point-based methods (Shi et al., 2019; Zhang et al., 2022b; Qi

et al., 2019; Fan et al., 2022; Yang et al., 2020) extract features for each LiDAR point. Voxel-based methods (Shi et al., 2020a; Yin et al., 2021; Yan et al., 2018; Lang et al., 2019) extract features on grid locations, resulting in a feature tensor. In this paper, we focus on point-based 3D object detectors.

**SSL on point clouds by contrastive learning.** Self-supervised learning (SSL) seeks to build feature representations without human annotations. The main branch of work is based on contrastive learning (Chen et al., 2020a; Chen & He, 2021; He et al., 2020b; Chen et al., 2020b; Grill et al., 2020; Henaff, 2020; Tian et al., 2020). For example, Zhang et al. (2021); Huang et al. (2021) treat each point cloud as an instance and learn the representation using contrastive learning. Liang et al. (2021); Xie et al. (2020); Yin et al. (2022) build point features by contrasting them across different regions in different scenes. The challenge is how to construct data pairs to contrast, which often needs extra information or ad hoc design such as spatial-temporal information (Huang et al., 2021) and ground planes (Yin et al., 2022). Our approach goes beyond contrastive learning for SSL on point clouds.

**SSL on point clouds by reconstruction.** Inspired by natural languages processing (Devlin et al., 2018; Raffel et al., 2020) and image understanding (Zhou et al., 2021; Assran et al., 2022; Larsson et al., 2017; Pathak et al., 2016), this line of work seeks to build feature representations by filling up missing data components (Wang et al., 2021; Pang et al., 2022; Yu et al., 2022; Liu et al., 2022; Zhang et al., 2022a; Eckart et al., 2021). However, most methods are limited to object-level reconstruction with synthetic data, *e.g.*, ShapeNet55 (Chang et al., 2015) and ModelNet40 (Wu et al., 2015). Only a few recent methods investigate outdoor driving scenes (Min et al., 2022; Lin & Wang, 2022), using the idea of masked autoencoders (MAE) (He et al., 2022). Our approach is related to reconstruction but with a fundamental difference — we do not reconstruct the input data (*i.e.*, LiDAR data) but the data from an associated modality (*i.e.*, color images). This enables us to bypass the mask design and leverage semantic cues in pixels, leading to features that facilitate 3D detection in outdoor scenes.

**Weak/Self-supervision from multiple modalities.** Learning representations from multiple modalities has been explored in different contexts. One prominent example is leveraging large-scale Internet text such as captions to supervise image representations (Radford et al., 2021; Singh et al., 2022; Driess et al., 2023). In this paper, we use images to supervise LiDAR-based representations. Several recent methods also use images but mainly to support contrastive learning (Sautier et al., 2022; Janda et al., 2023; Liu et al., 2021; 2020; Hou et al., 2021; Afham et al., 2022; Pang et al., 2023). Concretely, they form data pairs across modalities and learn two branches of feature extractors (one for each modality) to pull positive data pairs closer and push negative data pairs apart. In contrast, our work takes a reconstruction/generation approach. Our architecture is a single branch without pair construction.

**Colorization.** Image colorization has been an active research area in computer vision (Levin et al., 2004; Zhang et al., 2016), predicting colorful versions of grayscale images. In our paper, we introduce colorization as the pre-training task to equip the point cloud representation with semantic cues useful for the downstream 3D LiDAR-based detectors. We provide more reviews in Appendix S1.

## 3 PROPOSED APPROACH: GROUNDED POINT COLORIZATION (GPC)

Mainstream 3D object detectors take LiDAR point clouds as input. To localize and categorize objects accurately, the detector must produce an internal feature representation that can properly encode the semantic relationship among points. In this section, we introduce Grounded Point Colorization (GPC), which leverages color images as supervised signals to pre-train the detector's feature extractor (*i.e.*, backbone), enabling it to output semantically meaningful features without human supervision.

### 3.1 PRE-TRAINING VIA LEARNING TO COLORIZE

Color images can be easily obtained in autonomous driving scenarios since most sensor suites involve color cameras. With well-calibrated and synchronized cameras and LiDAR sensors, we can get the color of a LiDAR point by projecting it to the 2D image coordinate. As colors tend to be similar within an object instance but sharp across object boundaries, we postulate that by learning to colorize the point cloud, the resulting backbone would be equipped with the knowledge of objects.

**The first attempt.** Let us denote a point cloud by $\boldsymbol{X} = [\boldsymbol{x}_1, \cdots, \boldsymbol{x}_N] \in \mathbb{R}^{3 \times N}$; the corresponding color information by $\boldsymbol{Y} = [\boldsymbol{y}_1, \cdots, \boldsymbol{y}_N] \in \mathbb{R}^{3 \times N}$. The three dimensions in point $\boldsymbol{x}_n$ encode the 3D location in the ego-car coordinate system; the three dimensions in $\boldsymbol{y}_n$ encode the corresponding RGB pixel values. With this information, we can frame the pre-training of a detector backbone as a supervised colorization problem, treating each pair of $(\boldsymbol{X}, \boldsymbol{Y})$ as a labeled example.

Figure 2: **Architecture of GPC**. The key insight is grounding the pre-training colorization process on the hints, allowing the model backbone to focus on learning semantically meaningful representations that indicate which subsets (*i.e.*, segments) of points should be colored similarly to facilitate downstream 3D object detection.

Taking a point-based backbone $f_{\boldsymbol{\theta}}$ like PointNet++ (Qi et al., 2017b) as an example, the model takes $\boldsymbol{X}$ as input and outputs a feature matrix $\hat{\boldsymbol{Z}} \in \mathbb{R}^{D \times N}$, in which each column vector $\hat{\boldsymbol{z}}_n \in \mathbb{R}^D$ is meant to capture semantic cues for the corresponding point $\boldsymbol{x}_n$ to facilitate 3D object detection. One way to pre-train $f_{\boldsymbol{\theta}}$ using $(\boldsymbol{X}, \boldsymbol{Y})$ is to add another point-based model $g_{\boldsymbol{\phi}}$ (seen as the projection head or color decoder) on top of $f_{\boldsymbol{\theta}}$, which takes $\hat{\boldsymbol{Z}}$ as input and output $\hat{\boldsymbol{Y}}$ of the same shape as $\boldsymbol{Y}$. In this way, $\hat{\boldsymbol{Y}}$ can be compared with $\boldsymbol{Y}$ to derive the loss, for example, the Frobenius norm. Putting things together, we come up with the following optimization problem,

$$\min_{\boldsymbol{\theta}, \boldsymbol{\phi}} \sum_{(\boldsymbol{X}, \boldsymbol{Y}) \in \boldsymbol{D}_{\text{train}}} \|g_{\boldsymbol{\phi}} \circ f_{\boldsymbol{\theta}}(\boldsymbol{X}) - \boldsymbol{Y}\|_F^2 = \min_{\boldsymbol{\theta}, \boldsymbol{\phi}} \sum_{(\boldsymbol{X}, \boldsymbol{Y}) \in \boldsymbol{D}_{\text{train}}} \sum_n \|\hat{\boldsymbol{y}}_n - \boldsymbol{y}_n\|_2^2, \tag{1}$$

where $\boldsymbol{D}_{\text{train}}$ means the pre-training dataset and $\hat{\boldsymbol{y}}_n$ is the $n$-th column of $\hat{\boldsymbol{Y}} = g_{\boldsymbol{\phi}} \circ f_{\boldsymbol{\theta}}(\boldsymbol{X})$.

**The inherent color variation.** At first glance, Eq. 1 should be fairly straightforward to optimize using standard optimizers like stochastic gradient descent (SGD). In our experiments, we however found a drastically slow drop in training error, which results from the inherent color variation. That is, given $\boldsymbol{X}$, there is a huge variation of $\boldsymbol{Y}$ in the collected data due to environmental factors (*e.g.*, when the image is taken) or color options (*e.g.*, instances of the same object can have different colors). This implies a huge entropy in the conditional distribution $P(\boldsymbol{Y}|\boldsymbol{X})$, *i.e.*, a large $H(\boldsymbol{Y}|\boldsymbol{X}) = \mathbb{E}[-\log P(\boldsymbol{Y}|\boldsymbol{X})]$, indicating a small mutual information between $\boldsymbol{X}$ and $\boldsymbol{Y}$, *i.e.*, a small $I(\boldsymbol{X}; \boldsymbol{Y}) = H(\boldsymbol{Y}) - H(\boldsymbol{Y}|\boldsymbol{X})$. In other words, Eq. 1 may not provide sufficient information for $f_{\boldsymbol{\theta}}$ to learn.

Another way to look at this issue is the well-known bias-variance decomposition (Hastie et al., 2009; Bishop & Nasrabadi, 2006) in regression problems (suppose $\boldsymbol{Y} \in \mathbb{R}$), which contains a "noise" term,

$$\textbf{Noise: } \mathbb{E}_{\boldsymbol{Y}, \boldsymbol{X}} \left[ (\boldsymbol{Y} - \mathbb{E}_{\boldsymbol{Y}|\boldsymbol{X}}[\boldsymbol{Y}])^2 \right]. \tag{2}$$

When $P(\boldsymbol{Y}|\boldsymbol{X})$ has a high entropy, the noise term will be large, which explains the poor training error. Importantly, this noise term cannot be reduced solely by adopting a better model architecture. It can only be resolved by changing the data, or more specifically, how we leverage $(\boldsymbol{X}, \boldsymbol{Y})$.

One intuitive way to reduce $H(\boldsymbol{Y}|\boldsymbol{X})$ is to manipulate $P(\boldsymbol{Y}|\boldsymbol{X})$, *e.g.*, to carefully sub-sample data to reduce the variation. However, this would adversely reduce the amount of training data and also lead to a side effect of selection bias (Cawley & Talbot, 2010). In the extreme case, the pre-trained model may simply memorize the color of each distinct point cloud via the color decoder $g_{\boldsymbol{\phi}}$, preventing the backbone $f_{\boldsymbol{\theta}}$ from learning a useful embedding space for the downstream task.

**Grounded colorization with hints.** To resolve these issues, we propose to reduce the variance in $P(\boldsymbol{Y}|\boldsymbol{X})$ by providing additional conditioning evidence $\boldsymbol{S}$. From the perspective of information theory (Cover, 1999), adding evidence into $P(\boldsymbol{Y}|\boldsymbol{X})$, *i.e.*, $P(\boldsymbol{Y}|\boldsymbol{X}, \boldsymbol{S})$, guarantees,

$$H(\boldsymbol{Y}|\boldsymbol{X}) \geq H(\boldsymbol{Y}|\boldsymbol{X}, \boldsymbol{S}), \quad \text{and thus} \quad I((\boldsymbol{X}, \boldsymbol{S}); \boldsymbol{Y}) \geq I(\boldsymbol{X}; \boldsymbol{Y}), \tag{3}$$

unveiling more information for $f_{\boldsymbol{\theta}}$ to learn.

We realize this idea by proving ground-truth colors to a seed set of points in $\boldsymbol{X}$. Without loss of generality, we decompose $\boldsymbol{X}$ into two subsets $\boldsymbol{X}_S$ and $\boldsymbol{X}_U$ such that $\boldsymbol{X} = [\boldsymbol{X}_S, \boldsymbol{X}_U]$. Likewise, we decompose $\boldsymbol{Y}$ into $\boldsymbol{Y}_S$ and $\boldsymbol{Y}_U$ such that $\boldsymbol{Y} = [\boldsymbol{Y}_S, \boldsymbol{Y}_U]$. We then propose to inject $\boldsymbol{Y}_S$ as the conditioning evidence $\boldsymbol{S}$ into the colorization process. In other words, we directly provide the ground-truth colors to some of the points the model initially has to predict.

At first glance, this may controversially simplify the pre-training objective in Eq. 1. However, as the payback, we argue that this hint would help ground the colorization process on the inherent color

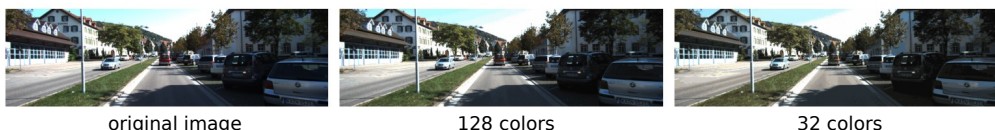

| original image | 128 colors | 32 colors |

Figure 3: **Color quantization.** We apply K-Means algorithm to cluster RGB pixel values into discrete bins. The resulting image with 128 bins is hardly distinguishable from the original image by human eyes.

variation, enabling the model to learn useful semantic cues for downstream 3D object detection. Concretely, to colorize a point cloud, the model needs to a) identify the segments that should possess similar colors and b) decide what colors to put on. With the hint that directly colors some of the points, the model can concentrate more on the first task, which aligns with the downstream task — to identify which subset of points to segment together.

**Sketch of implementation.** We implement our idea by injecting $\boldsymbol{Y}_S$ as additional input to the color decoder $g_\phi$. We achieve this by concatenating $\boldsymbol{Y}_S$ row-wise with the corresponding columns in $\hat{\boldsymbol{Z}}$, resulting in a $(D + 3)$-dimensional input to $g_\phi$. Let $\boldsymbol{O}_U$ denote an all-zero matrix with the same shape as $\boldsymbol{Y}_U$, we modify Eq. 1 as,

$$\min_{\boldsymbol{\theta}, \boldsymbol{\phi}} \sum_{(\boldsymbol{X}, \boldsymbol{Y}) \in D_{\text{train}}} \left\| g_\phi \left( \left[ \hat{\boldsymbol{Z}}; [\boldsymbol{Y}_S, \boldsymbol{O}_U] \right] \right) - [\boldsymbol{Y}_S, \boldsymbol{Y}_U] \right\|_F^2, \quad s.t. \quad \hat{\boldsymbol{Z}} = f_\theta \left( [\boldsymbol{X}_S, \boldsymbol{X}_U] \right), \quad (4)$$

where ";" denotes row-wise concatenation. Importantly, Eq. 4 only provides the hint $\boldsymbol{Y}_S$ to the color decoder $g_\phi$, leaving the input and output formats of the backbone model $f_\theta$ intact. We name our approach Grounded Point Colorization (GPC).

**GPC could mitigate selection bias.** Please see Appendix S2 for a discussion.

## 3.2 DETAILS OF GROUNDED POINT COLORIZATION (GPC)

**Model architecture.** For the backbone $f_\theta$, we use the model architectures defined in existing 3D object detectors. We specifically focus on detectors that use point-based feature extractors, such as PointRCNN (Shi et al., 2019) and IA-SSD (Zhang et al., 2022b). The output of $f_\theta$ is a set of point-wise features denoted by $\hat{\boldsymbol{Z}}$. For the color decoder $g_\phi$ that predicts the color of each input point, without loss of generality, we use a PointNet++ (Qi et al., 2017b).

**Color regression *vs*. color classification.** In Sec. 3.1, we frame colorization as a regression problem. However, regressing the RGB pixel values is sensitive to potential outlier pixels. In our implementation, we instead consider a classification problem by quantizing real RGB values into discrete bins. We apply the K-means algorithm over pixels of the pre-training images to cluster them into $K$ classes. The label of each LiDAR point $\boldsymbol{x}_n$ then becomes $y_n \in \{1, \cdots, K\}$. Fig. 3 shows the images with color quantization; each pixel is colored by the assigned cluster center. It is hard to distinguish the images with $K = 128$ bins from the original image by human eyes.

We accordingly modify the color decoder $g_\phi$. In the input, we concatenate a $K$-dimensional vector to the feature $\hat{\boldsymbol{z}}_n \in \mathbb{R}^D$ of each LiDAR point $\boldsymbol{x}_n$, resulting in a $(D + K)$-dimension input to $g_\phi$. For seed points where we inject ground-truth colors, the $K$-dimensional vector is *one-hot*, whose $y_n$-th element is one and zero elsewhere. For other LiDAR points, the $K$-dimensional vector is a zero vector. In the output, $g_\phi$ generates a $K$-dimensional logit vector $\hat{\boldsymbol{\eta}}_n \in \mathbb{R}^K$ for each input point $\boldsymbol{x}_n$. The predicted color class $\hat{y}_n$ can be obtained by $\arg\max_{c \in \{1, \cdots, K\}} \hat{\eta}_n[c]$.

**Pre-training objective.** The cross-entropy loss is the standard loss for classification. However, in driving scenes, most of the LiDAR points are on the ground, leading to an imbalanced distribution across color classes. We thus apply the Balanced Softmax (Ren et al., 2020) to re-weight each class,

$$\ell(\boldsymbol{x}_n, y_n) = -\log \left( \frac{\alpha_{y_n} \times \mathrm{e}^{\hat{\eta}_n[y_n]}}{\sum_{c=1}^K \alpha_c \times \mathrm{e}^{\hat{\eta}_n[c]}} \right), \quad (5)$$

where $\alpha_c$ is the balance factor of class $c$. In the original paper, $\alpha_c$ is proportional to the number of training examples of class $c$. Here, we set $\alpha_c$ per mini-batch, proportional to the number of points with class label $c$. A small $\epsilon$ is added to $\alpha_c$ for stability since an image may not contain all colors.

**Optimization.** In the pre-training stage, we train $f_\theta$ and $g_\phi$ end-to-end without human supervision. In the fine-tuning stage with labeled 3D object detection data, $g_\phi$ is disregarded; the pre-trained $f_\theta$ is fine-tuned end-to-end with other components in the 3D object detector.

Table 1: **Pre-trained on KITTI and fine-tuned on subsets of KITTI.** GPC is consistently better than scratch and PointContrast (Xie et al., 2020) especially when labeled data are extremely sparse.

| Sub. | Method | Detector | Car | | | Pedestrian | | | Cyclist | | | Overall | | |
|---|---|---|---|---|---|---|---|---|---|---|---|---|---|---|
| | | | easy | mod. | hard | easy | mod. | hard | easy | mod. | hard | easy | mod. | hard |
| 5% | scratch | PointRCNN | 66.8 | 51.6 | 45.0 | 61.5 | 55.4 | 48.3 | 81.4 | 58.6 | 54.5 | 69.9 | 55.2 | 49.3 |
| | PointContrast | PointRCNN | 58.2 | 46.7 | 40.1 | 52.5 | 47.5 | 42.0 | 73.1 | 49.7 | 46.6 | 61.2 | 47.9 | 42.9 |
| | GPC (Ours) | PointRCNN | 81.2 | 65.1 | 57.7 | 63.7 | **57.4** | 50.3 | 88.2 | **65.7** | 61.4 | 77.7 | **62.7** | 56.7 |
| | scratch | IA-SSD | 73.7 | 61.6 | 55.3 | 41.2 | 32.0 | 29.2 | 75.1 | 54.2 | 51.1 | 63.3 | 49.3 | 45.2 |
| | GPC (Ours) | IA-SSD | 81.3 | **66.6** | 59.7 | 49.2 | 40.5 | 36.7 | 77.3 | 54.7 | 51.2 | 69.3 | 53.9 | 49.2 |
| 20% | scratch | PointRCNN | 87.6 | 74.2 | 71.1 | 68.2 | 60.3 | 52.9 | 93.0 | 71.8 | 67.2 | 82.9 | 68.8 | 63.7 |
| | PointContrast | PointRCNN | 88.4 | 73.9 | 70.7 | 71.0 | **63.6** | 56.3 | 86.8 | 69.1 | 64.1 | 82.1 | 68.9 | 63.7 |
| | GPC (Ours) | PointRCNN | 88.7 | 76.0 | 72.0 | 70.6 | 63.0 | 55.8 | 92.6 | **73.0** | 68.1 | 84.0 | **70.7** | 65.3 |
| | scratch | IA-SSD | 86.1 | 74.0 | 70.6 | 52.2 | 45.6 | 40.1 | 84.5 | 64.0 | 59.8 | 74.3 | 61.2 | 56.8 |
| | GPC (Ours) | IA-SSD | 87.6 | **76.4** | 71.5 | 54.1 | 47.3 | 42.0 | 86.2 | 69.5 | 64.9 | 76.0 | 64.4 | 59.5 |
| 50% | scratch | PointRCNN | 89.7 | 77.4 | 74.9 | 68.9 | 61.3 | 54.2 | 88.4 | 69.2 | 64.4 | 82.3 | 69.3 | 64.5 |
| | PointContrast | PointRCNN | 89.1 | 77.3 | 74.8 | 69.2 | **62.0** | 55.1 | 90.4 | 70.8 | 66.5 | 82.9 | 70.0 | 65.4 |
| | GPC (Ours) | PointRCNN | 89.3 | 77.5 | 75.0 | 68.4 | 61.4 | 54.7 | 92.1 | 72.1 | 67.6 | 83.3 | **70.3** | 65.8 |
| | scratch | IA-SSD | 88.6 | 77.6 | 74.2 | 61.8 | 55.0 | 49.0 | 89.9 | 70.0 | 65.6 | 80.1 | 67.5 | 63.0 |
| | GPC (Ours) | IA-SSD | 89.3 | **80.1** | 76.9 | 62.5 | 55.4 | 49.6 | 88.8 | **73.7** | 69.2 | 80.2 | 69.7 | 65.2 |
| 100% | scratch | PointRCNN | 90.8 | 79.9 | 77.7 | 68.1 | 61.1 | 54.2 | 87.3 | 68.1 | 63.7 | 82.0 | 69.7 | 65.2 |
| | PointContrast | PointRCNN | 91.1 | 80.0 | 77.5 | 66.9 | 61.1 | 54.7 | 91.3 | 70.7 | 66.4 | 83.1 | 70.6 | 66.2 |
| | GPC (Ours) | PointRCNN | 90.8 | 79.6 | 77.4 | 68.5 | **61.3** | 54.7 | 92.3 | **71.8** | 67.1 | 83.9 | **70.9** | 66.4 |
| | scratch | IA-SSD | 89.3 | 80.5 | 77.4 | 60.1 | 52.9 | 46.7 | 87.8 | 69.7 | 65.2 | 79.1 | 67.7 | 63.1 |
| | GPC (Ours) | IA-SSD | 89.3 | **82.5** | 79.6 | 60.2 | 54.9 | 49.0 | 88.6 | 71.3 | 67.2 | 79.3 | 69.6 | 65.3 |

**All pieces together.** We propose a novel self-supervised learning algorithm GPC, which leverages colors to pre-train LiDAR-based 3D detectors. The model includes a backbone $f_\theta$ and a color decoder $g_\phi$; $f_\theta$ learns to generate embeddings to segment LiDAR points while $g_\phi$ learns to colorize LiDAR points. To mitigate the inherent color variation, GPC injects hints into $g_\phi$. To overcome the potential outlier pixels, GPC quantizes colors into class labels and treats colorization as a classification problem. Fig. 1 shows our main insight and Fig. 2 shows the architecture.

**Extension to voxel-based detectors.** As discussed in Sec. 2, some LiDAR-based detectors encode LiDAR points into a voxel-based feature representation. We investigate two ways of extension to voxel-based detectors. First, recent advancements, such as PV-RCNN (Shi et al., 2020a), have incorporated a point-based branch alongside the voxel-based backbone. This design allows GPC to still be applicable, as the color decoder $g_\phi$ can be applied to the point branch, enabling the pre-training signal to be backpropagated to the voxel-based backbone. The pre-trained backbone can then be utilized in any voxel-based detectors (see Sec. 4.2). Second, we explore enriching input data by augmenting LiDAR point coordinates with output features from the pre-trained point-based backbone $f_\theta$, which also yields promising results (see Appendix S4 for the results and discussion).

# 4 EXPERIMENTS

## 4.1 SETUP

**Datasets.** We use two commonly used datasets in autonomous driving: KITTI (Geiger et al., 2012) and Waymo Open Dataset (Sun et al., 2020). KITTI contains $7,481$ annotated samples, with $3,712$ for training and $3,769$ for validation. Waymo dataset is much larger, with $158,081$ frames in 798 scenes for training and $40,077$ frames in 202 scenes for validation. We pre-train on each dataset with all training examples without using any labels in the pre-training. We then fine-tune on different amounts of KITTI data, $5\%$, $10\%$, $20\%$, $50\%$, and $100\%$, with 3D annotations to investigate data-efficient learning. We adopt standard evaluation on KITTI validation set. Experiments of KITTI $\rightarrow$ KITTI show the in-domain performance. Waymo $\rightarrow$ KITTI provides the scalability and transfer ability of the algorithm, which is the general interest in SSL.

**3D Detectors.** As mentioned in Sec. 3, we focus on learning effective point representation with PointRCNN and IA-SSD. We see consistent improvement on both (Sec. 4.2). Additionally, we extend GPC to Voxel-based detectors as discussed in Sec. 3.2, Sec. 4.2, and Appendix S4.

**Implementation.** We sample images from the training set (3k for KITTI and 10k for Waymo) and 1k pixels for each image to learn the quantization. K-Means algorithm is adopted to cluster the colors into 128 discrete labels which are used for seeds and ground truths. The color labels in one hot concatenated with the features from the backbone are masked out $80\%$ of colors (*i.e.*, assigned to zeros). For 3D point augmentation, we follow the standard procedure, random flipping along

Table 2: **Pre-trained on Waymo and fine-tuned on subsets of KITTI.** We compare with DepthContrast (Zhang et al., 2021) and ProposalContrast (Yin et al., 2022). [†]directly taken from Yin et al. (2022).

| Sub. | Method | Detector | Car | | | Pedestrian | | | Cyclist | | | Overall | | |
|---|---|---|---|---|---|---|---|---|---|---|---|---|---|---|
| | | | easy | mod. | hard | easy | mod. | hard | easy | mod. | hard | easy | mod. | hard |
| 5% | scratch | PointRCNN | 66.8 | 51.6 | 45.0 | 61.5 | 55.4 | 48.3 | 81.4 | 58.6 | 54.5 | 69.9 | 55.2 | 49.3 |
| | GPC (Ours) | PointRCNN | 83.9 | **67.8** | 62.5 | 64.6 | **58.1** | 51.5 | 88.3 | **65.9** | 61.6 | 78.9 | **63.9** | 58.5 |
| | scratch | IA-SSD | 73.7 | 61.6 | 55.3 | 41.2 | 32.0 | 29.2 | 75.1 | 54.2 | 51.1 | 63.3 | 49.3 | 45.2 |
| | GPC (Ours) | IA-SSD | 81.5 | 67.4 | 60.1 | 47.2 | 39.5 | 34.5 | 81.2 | 61.4 | 57.4 | 70.0 | 56.1 | 50.7 |
| 20% | scratch[†] | PointRCNN | 88.6 | 75.2 | 72.5 | 55.5 | 48.9 | 42.2 | 85.4 | 66.4 | 71.7 | | 63.5 | |
| | ProposalContrast[†] | PointRCNN | 88.5 | **77.0** | 72.6 | 58.7 | 51.9 | 45.0 | 90.3 | 69.7 | 65.1 | | 66.2 | |
| | GPC (Ours) | PointRCNN | 90.9 | 76.6 | 73.5 | 70.5 | **62.8** | 55.5 | 89.9 | **71.0** | 66.6 | 83.8 | **70.1** | 65.2 |
| | scratch | IA-SSD | 86.1 | 74.0 | 70.6 | 52.2 | 45.6 | 40.1 | 84.5 | 64.0 | 59.8 | 74.3 | 61.2 | 56.8 |
| | GPC (Ours) | IA-SSD | 88.1 | 76.8 | 71.9 | 58.2 | 50.9 | 44.6 | 86.1 | 66.2 | 62.1 | 77.4 | 64.6 | 59.5 |
| 50% | scratch[†] | PointRCNN | 89.1 | 77.9 | 75.4 | 61.8 | 54.6 | 47.9 | 86.3 | 67.8 | 63.3 | | 66.7 | |
| | ProposalContrast[†] | PointRCNN | 89.3 | **80.0** | 77.4 | 62.2 | 54.5 | 46.5 | 92.3 | **73.3** | 68.5 | | 69.2 | |
| | GPC (Ours) | PointRCNN | 89.5 | 79.2 | 77.0 | 69.3 | **61.8** | 54.6 | 89.1 | 71.3 | 66.4 | 82.7 | **70.7** | 66.0 |
| | scratch | IA-SSD | 88.6 | 77.6 | 74.2 | 61.8 | 55.0 | 49.0 | 89.9 | 70.0 | 65.6 | 80.1 | 67.5 | 63.0 |
| | GPC (Ours) | IA-SSD | 88.4 | 79.8 | 76.4 | 63.4 | 56.2 | 49.6 | 88.3 | 70.5 | 66.0 | 80.1 | 68.9 | 64.0 |
| 100% | scratch[†] | PointRCNN | 90.0 | 80.6 | 78.0 | 62.6 | 55.7 | 48.7 | 89.9 | 72.1 | 67.5 | | 69.5 | |
| | DepthContrast[†] | PointRCNN | 89.4 | 80.3 | 77.9 | 65.6 | 57.6 | 51.0 | 90.5 | 72.8 | 68.2 | | 70.3 | |
| | ProposalContrast[†] | PointRCNN | 89.5 | 80.2 | 78.0 | 66.2 | 58.8 | 52.0 | 91.3 | **73.1** | 68.5 | | 70.7 | |
| | GPC (Ours) | PointRCNN | 89.1 | 79.5 | 77.2 | 67.9 | **61.5** | 55.0 | 89.1 | 71.0 | 67.0 | 82.0 | 70.7 | 66.4 |
| | scratch | IA-SSD | 89.3 | 80.5 | 77.4 | 60.1 | 52.9 | 46.7 | 87.8 | 69.7 | 65.2 | 79.1 | 67.7 | 63.1 |
| | GPC (Ours) | IA-SSD | 90.7 | **82.4** | 79.5 | 60.7 | 55.0 | 49.5 | 86.5 | 70.5 | 67.1 | 79.3 | 69.3 | 65.4 |

Table 3: **GPC on PV-RCNN (Shi et al., 2020a) and SOTA comparison.** The result shows that GPC can be applied on voxel-based detectors and it consistently performs better than existing methods.

| Sub. | Method | Car | Ped. | Cyc. | Overall |
|---|---|---|---|---|---|
| 20% | scratch | 82.52 | 53.33 | 64.28 | 66.71 |
| | ProposalContrast (Yin et al., 2022) | **82.65** | 55.05 | 66.68 | 68.13 |
| | PatchContrast (Shrout et al., 2023) | 82.63 | 57.77 | **71.84** | 70.75 |
| | GPC (Ours) | **82.65** | **59.87** | 71.59 | **71.37** |
| 50% | scratch | 82.68 | 57.10 | 69.12 | 69.63 |
| | ProposalContrast (Yin et al., 2022) | 82.92 | 59.92 | 72.45 | 71.76 |
| | PatchContrast (Shrout et al., 2023) | 84.47 | 60.76 | 71.94 | 72.39 |
| | GPC (Ours) | **84.58** | **61.06** | **72.61** | **72.75** |
| 100% | scratch | 84.50 | 57.06 | 70.14 | 70.57 |
| | GCC-3D (Liang et al., 2021) | - | - | - | 71.26 |
| | STRL (Huang et al., 2021) | 84.70 | 57.80 | 71.88 | 71.46 |
| | PointContrast (Xie et al., 2020) | 84.18 | 57.74 | 72.72 | 71.55 |
| | ProposalContrast (Yin et al., 2022) | **84.72** | 60.36 | 73.69 | 72.92 |
| | ALSO (Boulch et al., 2023) | 84.68 | 60.16 | 74.04 | 72.96 |
| | PatchContrast (Shrout et al., 2023) | 84.67 | 59.92 | **74.33** | 72.97 |
| | GPC (Ours) | 84.68 | **62.06** | 73.99 | **73.58** |

the X-axis, random rotation ($[-4/\pi, 4/\pi]$), random scaling ($[0.95, 1.05]$), random point sampling ($N = 16384$ with range$= 40$), and random point shuffling. We pre-train the model for 80 and 12 epochs on KITTI and Waymo respectively with batch sizes 16 and 32. We fine-tune on subsets of KITTI for 80 epochs with a batch size of 16 and the same 3D point augmentation above. The AdamW optimizer (Loshchilov & Hutter, 2017) and Cosine learning rate scheduler(Loshchilov & Hutter, 2016) are adopted in both stages which the maximum learning rate is 0.001 and 0.0004 respectively.

## 4.2 RESULTS

**Results on KITTI → KITTI.** We first study in-domain data-efficient learning with different amounts of labeled data available. Our results outperform random initialization (scratch) by a large margin, particularly when dealing with limited human-annotated data, as shown in Tab. 1. For instance, with PointRCNN, on 5% KITTI, we improve +7.5% on overall moderate AP (55.2 to 62.7). Remarkably, even with just 20% KITTI, we already surpass the performance achieved by training on the entire dataset from scratch, reaching a performance of 70.7. This highlights the benefit of GPC in data-efficient learning, especially when acquiring 3D annotations is expensive. Moreover, even when fine-tuning on 100% of the data, we still achieve a notable gain, improving from 69.7 to 70.9 (+1.2). We re-implement a recent method, PointContrast (Xie et al., 2020), as a stronger baseline. GPC performs better across all subsets of data. Notice that PointContrast struggles when annotations are extremely scarce, as seen on 5% subset where it underperforms the baseline (47.9 vs.55.2).

**Results on Waymo → KITTI.** We investigate the transfer learning ability on GPC as it's the general interest in SSL. We pre-train on Waymo (20× larger than KITTI) and fine-tune on subsets

Table 4: **Comparison with methods using images.** PPKT (Liu et al., 2021), SLidR (Sautier et al., 2022), and TriCC (Pang et al., 2023) pre-train on NuScenes (Caesar et al., 2020) with UNet (Choy et al., 2019). The results are shown in moderate AP. [†]directly taken from Pang et al. (2023).

| Method | Arch | Pre. Dataset | 5% | 10% | 20% |
|---|---|---|---|---|---|
| scratch[†] | UNet + PointRCNN | - | 56.6 | 58.8 | 63.7 |
| PPKT[†] | UNet + PointRCNN | NuScenes | 59.6 (**+3.0**) | 63.7 (**+4.9**) | 64.8 (**+1.1**) |
| SLidR[†] | UNet + PointRCNN | NuScenes | 58.8 (**+2.2**) | 63.5 (**+4.7**) | 63.8 (**+0.1**) |
| TriCC[†] | UNet + PointRCNN | NuScenes | 61.3 (**+4.7**) | 64.6 (**+5.8**) | 65.9 (**+2.2**) |
| scrach | PointRCNN | - | 55.2 | 65.2 | 68.8 |
| GPC (Ours) | PointRCNN | KITTI | 62.7 (**+7.5**) | 66.4 (**+1.2**) | 70.7 (**+1.9**) |
| GPC (Ours) | PointRCNN | Waymo | 63.9 (**+8.7**) | 67.0 (**+1.8**) | 70.1 (**+1.3**) |

Table 5: **Loss ablation for colorization.** We explore two regression losses Mean Squared Error (MSE), SmoothL1 (SL1) and two classification losses CrossEntropy (CE), modified Balanced Softmax (BS). The results are on KITTI $\rightarrow$ 5% KITTI. The best result is reported for each loss after a hyperparameter search.

| Loss | Type | Car | | | Pedestrian | | | Cyclist | | | Overall | | |
|---|---|---|---|---|---|---|---|---|---|---|---|---|---|
| | | easy | mod. | hard | easy | mod. | hard | easy | mod. | hard | easy | mod. | hard |
| scratch | - | 66.8 | 51.6 | 45.0 | 61.5 | 55.4 | 48.3 | 81.4 | 58.6 | 54.5 | 69.9 | 55.2 | 49.3 |
| MSE | reg | 72.4 | 59.8 | 52.9 | 64.2 | 56.2 | 49.2 | 86.2 | 64.2 | 60.2 | 74.2 | 60.1 | 54.1 |
| SL1 | reg | 76.8 | 62.9 | 56.0 | 63.2 | 55.6 | 49.2 | 84.1 | 63.3 | 58.9 | 74.7 | 60.6 | 54.7 |
| CE | cls | 78.4 | 64.5 | 58.5 | 60.7 | 54.1 | 47.6 | 85.5 | 61.5 | 57.6 | 74.9 | 60.0 | 54.6 |
| BS | cls | 81.2 | **65.1** | 57.7 | 63.7 | **57.4** | 50.3 | 88.2 | **65.7** | 61.4 | 77.7 | **62.7** | 56.5 |

of KITTI. In Tab. 2, our method shows consistent improvement upon scratch baseline. The overall moderate AP improves $55.2 \rightarrow 63.9$ on 5% KITTI. On 20%, it improves $63.5 \rightarrow 70.1$ which is again already better than 100% from scratch (69.5). Compared to the current state-of-the-art (SOTA), ProposalContrast (Yin et al., 2022), GPC outperforms on 20% (66.2 *vs.* 70.1) and 50% (69.2 *vs.* 70.7).

**Extension to Voxel-Based Detectors.** We extend the applicability of GPC to PV-RCNN (Shi et al., 2020a) for Waymo $\rightarrow$ KITTI. We randomly select $16,384$ key points, whose features are interpolated from voxelized features, for colorization supervision (see Sec. 3.2). Note that the learned voxelization backbone can be applied to any voxel-based detectors, not limited to PV-RCNN. As shown in Tab. 3, GPC consistently outperforms existing algorithms. The existing state-of-the-art (SOTA) algorithms primarily utilize contrastive learning which requires specific designs to identify meaningful regions to contrast. GPC, in contrast, offers a fresh perspective on pre-training 3D detectors. It is straightforward in its approach yet remarkably effective. Please see more results and discussion in Appendix S4.

**Learning with Images.** We've noticed recent efforts in self-supervised learning using images (see Sec. 2), where most approaches use separate feature extractors for LiDAR and images to establish correspondences for contrastive learning. In contrast, GPC directly utilizes colors within a single trainable architecture. We strive to incorporate numerous baselines for comparison, but achieving a fair comparison is challenging due to differences in data processing, model architecture, and training details. Therefore, we consider the baselines as references that demonstrate our approach's potential, rather than claiming SOTA status. In Tab. 4, we compare to PPKT (Liu et al., 2021), SLidR (Sautier et al., 2022) and TriCC (Pang et al., 2023) with moderate AP reported in Pang et al. (2023). Notably, on 5% KITTI, we outperform the baselines with substantial gains of +7.5 and +8.7, which are significantly higher than their improvements (less than +4.7).

## 4.3 Ablation Study

**Colorization with regression or classification?** In Tab. 5, we explore four losses for colorization. There is nearly no performance difference in using MSE, SL1, and CE, but BS outperforms other losses by a clear margin. We hypothesize that it prevents the learning bias toward the ground (major pixels) and forces the model to learn better on foreground objects that have much fewer pixels which facilitates the downstream 3D detection as well.

**Hint mechanism** is the key design to overcome the inherent color variation (see Sec. 3). In Tab. 7, we directly regress RGB values (row 2 and 3) without hints and color decoders. The performance is better than scratch but limited. We hypothesize that the naive regression is sensitive to pixel values and can't learn well if there exists a large color variation. The hint resolves the above issue and performs much better (row 4, 62.7 and row 5, 63.9). In Tab. 6, we ablate different levels of hint and find out providing only 20% as seeds is the best (61.9), suggesting the task to learn can't be too easy

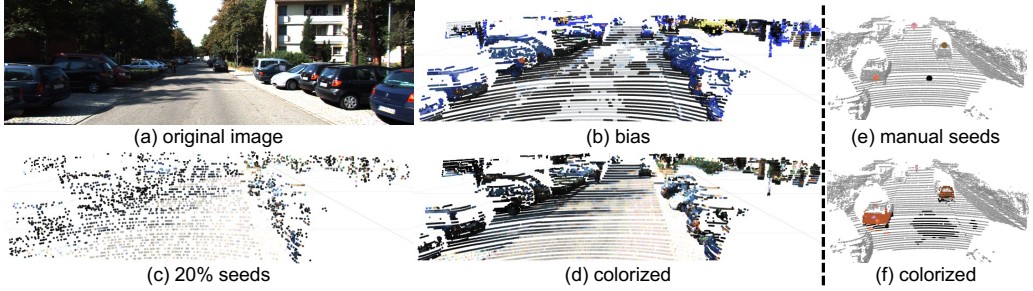

|  |  |  |
|---|---|---|
| (a) original image | (b) bias | (e) manual seeds |
| (c) 20% seeds | (d) colorized | (f) colorized |

Figure 4: **Qualitative results of colorization.** (left) We use trained GPC to infer on validation LiDAR (b) without any seeds and with (c) resulting in (d). The model learned the bias about the driving scene: the ground is gray and black, trees are green, and tail lights are red, but it predicts the average color on all cars because they can be variant. On the other hand, it correctly colorizes the point cloud if some hints are provided. We hypothesize such an ability to know where to pass the color is essential for downstream 3D detection. (right) (e) We further manipulate the seeds by hand. (f) GPC successfully colorize the regions with given colors in (e).

Table 6: **Seed ratios.**

| seed | easy | mod. | hard |
|---|---|---|---|
| 100% | 51.7 | 38.7 | 33.8 |
| 80% | 66.4 | 50.5 | 44.9 |
| 40% | 75.1 | 59.7 | 53.4 |
| 20% | 77.1 | 61.9 | 56.2 |
| 0% | 57.9 | 44.1 | 38.4 |
| scratch | 69.9 | 55.2 | 49.3 |

Table 7: **Hint mechanism**: with and without color decoder.

| Hint | Pre. Dataset | easy | mod. | hard |
|---|---|---|---|---|
| scratch | - | 69.9 | 55.2 | 49.3 |
|  | KITTI | 75.1 | 59.4 | 53.3 |
|  | Waymo | 74.9 | 60.4 | 54.5 |
| ✓ | KITTI | 77.7 | 62.7 | 56.5 |
| ✓ | Waymo | 78.9 | 63.9 | 58.5 |

Table 8: **Number of epochs pertaining on Waymo.**

| epochs | easy | mod. | hard |
|---|---|---|---|
| 1 | 75.8 | 60.7 | 54.4 |
| 4 | 76.4 | 61.6 | 55.9 |
| 8 | 78.3 | 63.1 | 58.4 |
| 12 | 78.9 | 63.9 | 58.5 |

(*i.e.*, providing too many hints). However, providing no hints falls into the issue of color variation again, deviating from learning a good representation (0%, 44.1 worse than scratch, 55.2).

**Repeated experiments.** To address potential performance fluctuations with different data sampled for fine-tuning, we repeat the 5% fine-tuning experiment for another five rounds, each with a distinctly sampled subset, as shown in Appendix S5. The results show the consistency and promise of GPC.

## 4.4 QUALITATIVE RESULTS

We validate GPC by visualization as shown in Fig. 4 (left). Impressively the model can learn common sense about colors in the driving scene in (b), even red tail lights on cars. We further play with manual seeds to manipulate the object colors (Fig. 4 (right)) and it more or less recognizes which points should have the same color and stops passing to the ground.

## 4.5 DISCUSSION

GPC outperforms the mainstream contrastive learning-based methods, suggesting our new perspective as a promising direction to explore further. It is particularly advantageous when the labeled data are scarce. Regardless of the 3D model and the pre-training dataset, GPC 's performance of fine-tuning on just 20% of labeled data is consistently better than training from scratch with the entire dataset (Tab. 1, Tab. 2, Tab. 3). This advantage holds the potential to reduce the annotation effort required.

## 5 CONCLUSION

Color images collected along with LiDAR data offer valuable semantic cues about objects. We leverage this insight to frame pre-training LiDAR-based 3D object detectors as a supervised colorization problem, a novel perspective drastically different from the mainstream contrastive learning-based approaches. Our proposed approach, Grounded Point Colorization (GPC), effectively addresses the inherent variation in the ground-truth colors, enabling the model backbone to learn to identify LiDAR points belonging to the same objects. As a result, GPC could notably facilitate downstream 3D object detection, especially under label-scarce settings, as evidenced by extensive experiments on benchmark datasets. We hope our work can vitalize the community with a new perspective.

## ACKNOWLEDGMENT

This research is supported in part by grants from the National Science Foundation (IIS-2107077, OAC-2118240, OAC-2112606, IIS-2107161, III-1526012, and IIS-1149882), NVIDIA research, and Cisco Research. We are thankful for the generous support of the computational resources by the Ohio Supercomputer Center. We also thank Mengdi Fan for her valuable assistance in creating the outstanding figures for this work.

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
