# Supplementary Material for
# Pre-training LiDAR-based 3D Object Detectors through Colorization

In this supplementary material, we provide more details and experiment results in addition to the main paper:

- Appendix S1: provides literature review about colorization.
- Appendix S2: discusses how GPC could mitigate selection bias.
- Appendix S3: provides more evidence to show the effectiveness of GPC.
- Appendix S4: discusses more details about extension to voxel-based detecros.
- Appendix S5: repeats the experiments to show consistent performance.
- Appendix S6: explores another way to sample points for colorization.
- Appendix S7: explores another usage of color.
- Appendix S8: provides more results on Waymo to Waymo setting.

## S1 COLORIZATION

Image colorization has been an active research area in computer vision (Zhang et al., 2016; Larsson et al., 2016; Su et al., 2020; Kumar et al., 2021; Weng et al., 2022; Ji et al., 2022; He et al., 2018; Chang et al., 2023; Levin et al., 2004; Kim et al., 2022; Xia et al., 2022; Larsson et al., 2017; Treneska et al., 2022), predicting colorful versions of grayscale images. Typically, this task relies on human-related cues such as scribbles (Levin et al., 2004), exemplars (He et al., 2018), or texts (Chang et al., 2023), as well as external priors like pre-trained detectors (Su et al., 2020), generative models (Kim et al., 2022), or superpixels (Xia et al., 2022; Wan et al., 2020), to achieve realistic and diverse colorization. Some works have extended the task to 3D point clouds (Cao & Nagao, 2019; Liu et al., 2019; Shinohara et al., 2021; Gao et al., 2023). Specifically, Treneska et al. (2022) and Shinohara et al. (2021) use a GAN-style loss, learning a discriminator to differentiate real colorful images and colorized images and using it to train the colorization model. This provides an alternative to using hints to resolve the color variation problem.

Instead of treating colorization as the ultimate task, Zhang et al. (2016), Larsson et al. (2017), and Treneska et al. (2022) have explored colorization as a pre-training task for 2D detection and segmentation. We note that Zhang et al. (2016) and Larsson et al. (2017) do not specifically address the color variation problem.

In our paper, we introduce colorization as the pre-training task to equip the point cloud representation with semantic cues useful for the downstream 3D LiDAR-based detectors. We identify color variation as the intrinsic challenge and propose inserting color hints into the output of the backbone to ground the colorization taking place in the color decoder (cf. Fig. 2). We note that such a design is crucial—we cannot add color hints as input to the backbone as the downstream task takes only the LiDAR points as input. In contrast, for colorization methods not for pre-training, the hints can be directly added to the input gray-level image (Levin et al., 2004). Wan et al. (2020) proposed a hybrid method to first predict the color of each superpixel pixel using a neural network, and propagated them to nearby pixels using a hand-crafted method (e.g., 2D-PCA). In contrast, our GPC learns to propagate colors in an end-to-end fashion by directly minimizing the colorization loss, which is the key to benefit downstream 3D detection.

## S2 GROUNDED POINT COLORIZATION COULD MITIGATE SELECTION BIAS

Another advantage of injecting hints into the colorization process is its ability to overcome selection bias Cawley & Talbot (2010) — the bias that only a few variations of $Y$ conditioned on $X$ are collected in the dataset. According to Clark et al. (2019) and Cadene et al. (2019), one effective

approach to prevent the model from learning the bias is to proactively augment the bias to the "output" of the model during the training phase. The augmentation, by default, will reduce the loss on the training example that contains bias, allowing the model to focus on hard examples in the training set. We argue that our model design and objective function Eq. 4 share similar concepts with Clark et al. (2019) and Cadene et al. (2019).

Another way to mitigate bias in $Y$ is to perform data augmentation, *e.g.*, by color jittering. We note that this is not applicable through Eq. 1 as it will adversely increase the entropy $H(Y|X)$, but is applicable through Eq. 4 via grounded colorization.

## S3  EFFECTIVENESS OF GPC

In the main paper, we show the superior performance of our proposed pre-training method, Grounded Point Colorization (GPC) (Sec. 4.2). With our pre-trained backbone, it outperforms random initialization as well as existing methods on fine-tuning with different amounts of labeled data (*i.e.*5%, 20%, 50%, and 100%) of KITTI (Geiger et al., 2012). Significantly, fine-tuning on only 20% with GPC is already better than training 100% from scratch. Besides, GPC demonstrates the ability of transfer learning by pre-training on Waymo (Sun et al., 2020) in Tab. 2. Not to mention, GPC is fairly simpler than existing methods which require road planes for better region sampling, temporal information for pair construction to contrast, or additional feature extractors for different modalities (Sec. 2). Most importantly, We successfully explore a new perspective of the pre-training methods on LiDAR-based 3D detectors besides contrastive learning-based algorithms.

In this supplementary, we further provide the training curve besides the above evidence. Fig. S1 shows the evaluation on KITTI validation set and loss on the training set during the fine-tuning on 5% KITTI. With GPC, the overall moderate AP raises faster than the scratch baseline and keeps the gain till the end. It already performs better at 55th epoch than the final performance of scratch baseline, which is more than 30% better in terms of efficiency. The training loss with GPC is also below the curve of the scratch baseline.

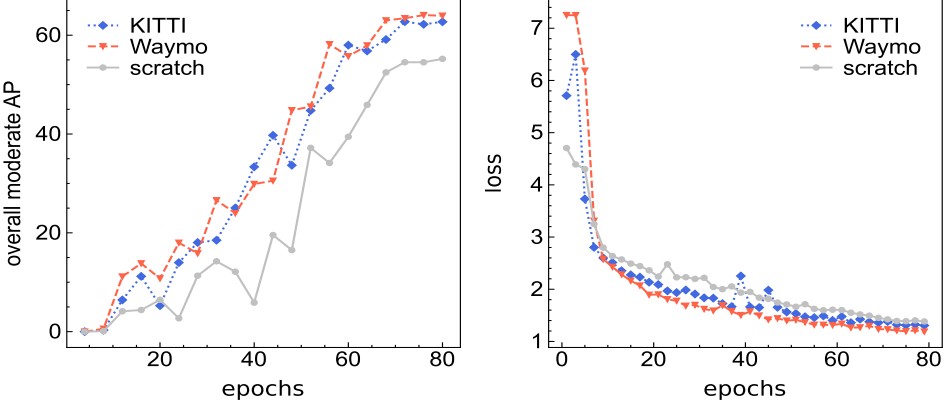

Figure S1: **Training curve on 5% KITTI.** (*left*) is the performance (↑) on KITTI validation set and (*right*) is the loss (↓) on the training set during the fine-tuning. It shows the effectiveness and efficiency of GPC compared to training from scratch.

With the above evidence and the extensive experiments in the main paper, we explain that GPC is a simple yet effective self-supervised learning algorithm for pre-training LiDAR-based 3D object detectors without human-labeled data.

## S4  EXTENSION TO VOXEL-BASED DETECTOR.

As discussed in Sec. 3.2, we investigate two ways to extend GPC to voxel-based detectors. The first approach is utilizing the recent architecture, such as PV-RCNN (Shi et al., 2020a), which contains both voxel-based and point-based branch. GPC provides self-supervision on the point-based branch which then is backpropagated to the voxelized backbone. The pre-trained backbone can be applied to any voxel-based detector. We show the full results in Tab. 3 of the main paper. GPC achieves state-of-the-art compared to other algorithms.

Table S1: **Approach with GPC on SECOND (Yan et al., 2018).** The result shows that the point representation by GPC can be a good plug-and-play in addition to the point coordinate (*i.e.* xyz) for better 3D detection.

| Sub. | Method | Pre. on | Car | | | Pedestrian | | | Cyclist | | | Overall | | |
|---|---|---|---|---|---|---|---|---|---|---|---|---|---|---|
| | | | easy | mod. | hard | easy | mod. | hard | easy | mod. | hard | easy | mod. | hard |
| 5% | scratch | - | 64.9 | 52.7 | 45.3 | 44.3 | 39.9 | 35.0 | 71.0 | 50.4 | 47.1 | 60.1 | 47.7 | 42.5 |
| | GPC (Ours) | KITTI | 74.9 | **60.8** | 54.1 | 46.8 | 41.9 | 37.3 | 74.2 | **55.0** | 51.5 | 65.3 | **52.6** | 47.6 |
| | GPC (Ours) | Waymo | 69.5 | 57.1 | 50.5 | 50.1 | **43.4** | 38.7 | 74.2 | 53.8 | 49.9 | 64.6 | 51.4 | 46.4 |
| 20% | scratch | - | 83.9 | 71.7 | 67.4 | 54.7 | 48.2 | 42.8 | 74.3 | 58.0 | 54.6 | 71.0 | 59.3 | 54.9 |
| | GPC (Ours) | KITTI | 85.6 | 72.0 | 67.4 | 54.2 | 47.6 | 42.4 | 78.2 | **60.9** | 57.1 | 72.7 | **60.2** | 55.6 |
| | GPC (Ours) | Waymo | 85.1 | **72.3** | 67.8 | 56.6 | **48.7** | 43.3 | 75.4 | 59.6 | 55.8 | 72.4 | **60.2** | 55.6 |
| 50% | scratch | - | 88.1 | 76.2 | 73.2 | 59.7 | 52.4 | 46.0 | 83.9 | 63.0 | 59.4 | 77.2 | 63.8 | 59.5 |
| | GPC (Ours) | KITTI | 87.5 | 75.8 | 73.3 | 60.3 | 52.5 | 46.0 | 78.3 | 61.1 | 57.5 | 75.4 | 63.1 | 58.9 |
| | GPC (Ours) | Waymo | 88.0 | **76.3** | 73.5 | 60.7 | **53.3** | 47.0 | 85.5 | **67.5** | 63.4 | 78.1 | **65.7** | 61.3 |
| 100% | scratch | - | 88.1 | 78.3 | 73.7 | 62.0 | 54.4 | 47.6 | 81.1 | 62.4 | 58.7 | 77.1 | 65.0 | 60.0 |
| | GPC (Ours) | KITTI | 88.7 | **79.2** | 74.5 | 63.4 | 56.0 | 48.8 | 84.8 | **64.5** | 60.6 | 79.0 | 66.5 | 61.3 |
| | GPC (Ours) | Waymo | 90.9 | 79.1 | 76.2 | 65.2 | **56.9** | 50.0 | 82.5 | **64.5** | 60.6 | 79.5 | **66.8** | 62.3 |

The second approach is concatenating our pre-trained per-point feature with the raw input (*i.e.*, the LiDAR point coordinate). We validate this approach on SECOND (Yan et al., 2018). More specifically, instead of just using the coordinate of points in each voxel, we concatenate xyz with the feature from the PointNet++ backbone pre-trained by GPC (mathematically $\boldsymbol{X}' = [\ \boldsymbol{X}\ |\ f_{\boldsymbol{\theta}}(\boldsymbol{X})\ ] \in \mathbb{R}^{(3+D) \times N}$). SECOND is randomly initialized and no projection layers are added. PointNet++ backbone is frozen during the fine-tuning. All the other training details are standard without hard tuning, including data augmentation, training scheduler, optimizer, *etc*.

The result is shown in Tab. S1. Impressively, even with this approach, GPC still brings significant improvement upon the scratch baseline. On $5\%$ KITTI, it improves from $47.7$ to $52.6$ on overall moderate AP, which is a gain of $+4.9$. On $50\%$ KITTI, the performance of pre-training on Waymo $(65.7)$ is already better than training with $100\%$ from scratch $(65.0)$. Even on $100\%$ KITTI, GPC still brings more than $+1.5$ gain upon scratch baseline. It demonstrates the effectiveness of our learned point representation and opens the great potential of our proposed method.

## S5 REPEATED EXPERIMENTS

To address potential performance fluctuations with different data sampled for fine-tuning, we repeat the $5\%$ fine-tuning experiment for another five rounds, each with a distinctly sampled subset. The results are summarized in Tab. S2, in which we compare the baseline of training from scratch (with $5\%$ KITTI data), pre-training with full KITTI (cf. Tab. 8 in the main paper), and pre-training with Waymo (cf. Tab. 2 in the main paper). Pre-training with our GPC consistently outperforms the baseline. Notably, the standard deviation among the five rounds is largely reduced with pre-training. For instance, the standard deviation of the overall moderate AP is 2.7 with the scratch baseline, which is reduced to 0.6 with GPC. This significant reduction in standard deviation underscores the consistency and promise of our proposed method.

## S6 POINT SAMPLING FOR COLORIZATION

In the main paper, the hints provided in the pre-training are completely random. We agree that, during pre-training, it may not be necessary to provide as many hints for points located on the ground, and instead, we should focus more on foreground points. One potential solution is to sample fewer points from the major color classes. For example, the number of sampled points per class is set proportional to the square root of the total number of points per class. We have implemented the idea and reported the results of fine-tuning on $5\%$ KITTI in Tab. S3. We found no significant difference with more balanced sampling. However, we agree that smarter sampling is indeed interesting for future research and exploration.

Table S2: **Repeat experiments.** We re-sample 5 different subsets of 5% KITTI and repeat the experiments. The standard deviation of overall moderate AP on the scratch baseline is 2.7, but it's only 0.6 on GPC. It implies that our improvement is promising and constant, regardless of the performance on the baseline.

| Exp. | Pre. | Car | | | Pedestrian | | | Cyclist | | | Overall | | |
|---|---|---|---|---|---|---|---|---|---|---|---|---|---|
| | | easy | mod. | hard | easy | mod. | hard | easy | mod. | hard | easy | mod. | hard |
| 1 | scratch | 51.3 | 41.9 | 36.2 | 56.9 | 51.4 | 45.0 | 74.5 | 53.7 | 50.0 | 60.9 | 49.0 | 43.7 |
| | KITTI | 79.2 | 65.5 | 59.1 | 62.7 | 55.5 | 48.4 | **87.9** | 65.0 | 61.0 | 76.6 | 62.0 | 56.1 |
| | Waymo | **83.8** | **68.4** | **63.1** | **65.6** | **58.7** | **51.4** | 87.1 | **65.9** | **61.4** | **78.8** | **64.3** | **58.7** |
| 2 | scratch | 44.0 | 36.5 | 31.6 | 54.6 | 48.8 | 43.0 | 70.9 | 48.7 | 45.5 | 56.5 | 44.7 | 40.1 |
| | KITTI | 81.1 | 65.0 | 58.8 | 60.3 | 53.7 | 47.4 | 86.8 | 63.4 | 59.4 | 76.1 | 60.7 | 55.2 |
| | Waymo | **82.2** | **67.6** | **62.4** | **64.2** | **57.3** | **50.9** | **87.0** | **65.3** | **61.5** | **77.8** | **63.4** | **58.3** |
| 3 | scratch | 63.0 | 49.1 | 42.1 | 57.6 | 51.3 | 44.8 | 80.6 | 58.0 | 53.8 | 67.1 | 52.8 | 46.9 |
| | KITTI | 78.3 | 64.0 | 56.9 | 61.4 | 54.4 | 47.3 | **86.5** | 62.6 | 58.5 | 75.4 | 60.3 | 54.2 |
| | Waymo | **80.2** | **64.5** | **59.5** | **65.9** | **57.6** | **50.7** | 86.4 | **66.1** | **61.8** | **77.5** | **62.7** | **57.3** |
| 4 | scratch | 59.5 | 46.3 | 39.2 | 52.9 | 47.9 | 42.0 | 72.4 | 51.1 | 47.7 | 61.6 | 48.4 | 43.0 |
| | KITTI | 82.5 | 67.4 | 60.1 | **64.9** | **57.0** | 49.7 | **87.6** | 65.4 | 61.1 | 78.3 | 63.3 | 57.0 |
| | Waymo | **85.1** | **68.4** | **63.1** | 63.5 | 56.7 | **50.0** | 87.4 | **66.3** | **62.1** | **78.7** | **63.8** | **58.4** |
| 5 | scratch | 52.6 | 42.8 | 36.8 | 51.8 | 47.3 | 42.0 | 71.2 | 50.0 | 46.6 | 58.5 | 46.7 | 41.8 |
| | KITTI | 81.1 | 65.2 | 58.9 | 60.7 | 53.8 | 46.9 | **89.8** | 65.7 | 61.7 | 77.2 | 61.6 | 55.9 |
| | Waymo | **84.5** | **68.4** | **64.9** | **65.0** | **57.3** | **51.3** | 87.0 | **66.4** | **62.5** | **78.8** | **64.0** | **59.6** |

Table S3: **Comparison of point sampling method.**

| Overall mod. | Car mod. | Ped. mod. | Cyc. mod. | Overall mod. |
|---|---|---|---|---|
| random sampling | 65.7 | 65.1 | 57.4 | 62.7 |
| more balanced sampling | 66.2 | 64.9 | 56.5 | 62.5 |

## S7 CORRECT USAGE OF COLOR

GPC uses color to provide semantic cues to pre-train LiDAR-based 3D detectors (*c.f.* Sec. 3 in the main paper). With extensive experiments (*c.f.* Sec. 4.2 and Sec. 4.3), we show that the hint mechanism is the key to learning good point representation. Naive regression of color without hints can hurt performance. In this supplementary, we further explore a possibility: directly concatenating the input coordinate (*i.e.* xyz) with pixel values (*i.e.* RGB) to form a 6-dimension input vector. As shown in Tab. S4, the performance is even worse than plain scratch. It implies that the additional color information on the input isn't necessary to obtain the richer representation. On the other hand, using it with GPC in the pre-training stage shows significant improvement. We argue that how to leverage color information correctly is not trivial. Also, notice that color is no longer needed during fine-tuning.

## S8 RESULTS ON WAYMO TO WAYMO.

We evaluate on 1% Waymo with its standard metric AP and APH. The result in Tab. S5 shows that GPC improve +2.4% on AP and +4.4% on APH.

Table S4: **Comparison of usages of color.** Directly concatenating the color (*i.e.*RGB) and the coordinate (*i.e.*xyz) performs worse than plan scratch on all subsets. In contrast, GPC utilizing the color as semantic cues in the pre-training brings significant improvement.

| Sub. | Method | Car | | | Pedestrian | | | Cyclist | | | Overall | | |
|---|---|---|---|---|---|---|---|---|---|---|---|---|---|
| | | easy | mod. | hard | easy | mod. | hard | easy | mod. | hard | easy | mod. | hard |
| 5% | scratch | 66.8 | 51.6 | 45.0 | 61.5 | 55.4 | 48.3 | 81.4 | 58.6 | 54.5 | 69.9 | 55.2 | 49.3 |
| | w/ color | 57.5 | 46.3 | 38.9 | 50.8 | 46.6 | 41.1 | 72.9 | 54.1 | 50.5 | 60.4 | 49.0 | 43.5 |
| | GPC | 81.2 | **65.1** | 57.7 | 63.7 | **57.4** | 50.3 | 88.2 | **65.7** | 61.4 | 77.7 | **62.7** | 56.7 |
| 20% | scratch | 87.6 | 74.2 | 71.1 | 68.2 | 60.3 | 52.9 | 93.0 | 71.8 | 67.2 | 82.9 | 68.8 | 63.7 |
| | w/ color | 87.4 | 73.6 | 70.2 | 60.4 | 54.8 | 48.6 | 90.0 | 69.9 | 65.3 | 79.2 | 66.1 | 61.4 |
| | GPC | 88.7 | **76.0** | 72.0 | 70.6 | **63.0** | 55.8 | 92.6 | **73.0** | 68.1 | 84.0 | **70.7** | 65.3 |
| 50% | scratch | 89.7 | 77.4 | 74.9 | 68.9 | 61.3 | 54.2 | 88.4 | 69.2 | 64.4 | 82.3 | 69.3 | 64.5 |
| | w/ color | 88.8 | 77.3 | 74.7 | 65.5 | 59.3 | 53.2 | 92.6 | 71.0 | 67.4 | 82.3 | 69.2 | 65.1 |
| | GPC | 89.3 | **77.5** | 75.0 | 68.4 | **61.4** | 54.7 | 92.1 | **72.1** | 67.6 | 83.3 | **70.3** | 65.8 |
| 100% | scratch | 90.8 | 79.9 | 77.7 | 68.1 | 61.1 | 54.2 | 87.3 | 68.1 | 63.7 | 82.0 | 69.7 | 65.2 |
| | w/ color | 90.8 | **80.0** | 77.5 | 63.2 | 57.0 | 51.4 | 89.5 | 69.7 | 65.1 | 81.2 | 68.9 | 64.7 |
| | GPC | 90.8 | 79.6 | 77.4 | 68.5 | **61.3** | 54.7 | 92.3 | **71.8** | 67.1 | 83.9 | **70.9** | 66.4 |

Table S5: **Pre-train and fine-tune on Waymo.** Performance of LEVEL 2 is reported.

| Sub. | Method | Detector | Vehicle | | Pedestrian | | Cyclist | | Overall | |
|---|---|---|---|---|---|---|---|---|---|---|
| | | | AP | APH | AP | APH | AP | APH | AP | APH |
| 1% | scratch | IA-SSD | 45.55 | 44.60 | 32.88 | 16.75 | 43.07 | 29.06 | 40.50 | 30.14 |
| | GPC (Ours) | IA-SSD | **48.25** | **47.53** | **37.09** | **23.90** | **43.42** | **35.30** | **42.92** | **35.58** |