# OpenReview forum: "Pre-training LiDAR-based 3D Object Detectors through Colorization"
_ICLR.cc/2024/Conference — ICLR 2024 poster_

### Official Review · Reviewer_anv1 · 2023-10-28

**Soundness:** 3 good
**Presentation:** 2 fair
**Contribution:** 2 fair
**Rating:** 6
**Confidence:** 5

**Summary:**

This paper proposes a new pretraining method for LiDAR-based 3D object detection, by colorization. In particular, to mitigate the ambiguity in colorization, this paper proposes Grounded Point Colorization (GPC), introducing some seed points as hints.
I believe the idea is good and makes sense to me.
Experiments are mainly conducted in the KITTI dataset, showing promising results but reaching the SOTA performance.

**Strengths:**

- The proposed method of Grounded Point Colorization (GPC) is reasonable. The use of seed points is also a good design.
- This paper is well-written and easy to follow.
- Though the experiments are mainly conducted in KITTI, they are thoughtful and complete.

**Weaknesses:**

- Authors use object detection as the downstream task, but I personally believe LiDAR-based segmentation is the best choice. Colorization-based pre-training mainly learns the semantics in my opinion, but object detection needs accurate locations and poses especially in the benchmark using IoU-based metrics such as KITTI and Waymo.
- The ultimate performance is not that good. The results on the Waymo Open Dataset are too weak. Other results on KITTI are also not convincing because PointRCNN and IA-SSD are not SOTA detectors nowadays.
- Although the writing is clear, there are some unnecessary parts like the decorative math on page 4, which do not make the paper better.

**Questions:**

- According to my first concern in the weaknesses box, I wonder why not authors adopt segmentation as the downstream task? Or some other detection benchmarks relying on accurate semantics such as nuScenes? I believe the proposed method can obtain better performance on these tasks.
- Authors should adopt SOTA detectors for experiments on KITTI, such as PVRCNN. The effectiveness should further be proven on Waymo Open Dataset with SOTA detectors such as PVRCNN, FSD, etc. The current performance is not competitive enough.

PV-RCNN: Point-Voxel Feature Set Abstraction for 3D Object Detection
FSD: Fully Sparse 3D Object Detection

---

> ### Author Response · Authors · 2023-11-23
> **Rebuttal (Part 1)**
>
> **Downstream tasks.**
> Thanks for the comment. In our humble opinion, our colorization-based pre-training (i.e., GPC) does not just learn semantics but which nearby points belong to the same instance (e.g., cf. the last paragraph on page 4). In essence, the same semantic class can have different colors, even in the same scene (e.g., multiple cars of different colors in Fig. 4). Our method can accurately colorize each car instance (cf. Fig. 4 (d)), implying that it learns the notion of object instances, which we believe crucial for 3D object detection.
>
> That said, during the rebuttal, we followed your suggestion to perform a preliminary study of GPC on LiDAR-point semantic segmentation. We use a UNet-like model as in PointContrast and perform experiments on NuScenes. While we did not have sufficient time to perform hyper-parameter tuning, we have already obtained a 6% gain on mIoU against training from scratch. We will include the comprehensive results in the final version.
>
> **Using SOTA detectors.**
> We provide additional results of pre-training on Waymo and fine-tuning on KITTI (cf. Table 2), using PV-RCNN as the 3D detector. (We note that many SOTA results use PV-RCNN.) As shown in the following table, GPC consistently outperforms existing algorithms. Please also see our response **More comparisons in Table 1 and Table 2** to Reviewer Hnni for further discussions.
>
> | Labels |      Method      | Overall mod. |  Car mod. | Ped. mod. | Cyc. mod. |
> |:------:|:----------------:|:------------:|:---------:|:---------:|:---------:|
> |   20%  | scratch          |     66.71    |   82.52   |   53.33   |   64.28   |
> |        | ProposalContrast |     68.13    | **82.65** |   55.05   |   66.68   |
> |        | PatchContrast    |     70.75    |   82.63   |   57.77   | **71.84** |
> |        | GPC (Ours)       |   **71.37**  | **82.65** | **59.87** |   71.59   |
> |   50%  | scratch          |     69.63    |   82.68   |   57.10   |   69.12   |
> |        | ProposalContrast |     71.76    |   82.92   |   59.92   |   72.45   |
> |        | PatchContrast    |     72.39    |   84.47   |   60.76   |   71.94   |
> |        | GPC (Ours)       |   **72.75**  | **84.58** | **61.06** | **72.61** |
> |  100%  | scratch          |     70.57    |   84.50   |   57.06   |   70.14   |
> |        | GCC-3D           |     71.26    |     -     |     -     |     -     |
> |        | STRL             |     71.46    |   84.70   |   57.80   |   71.88   |
> |        | PointContrast    |     71.55    |   84.18   |   57.74   |   72.72   |
> |        | ProposalContrast |     72.92    | **84.72** |   60.36   |   73.69   |
> |        | ALSO             |     72.96    |   84.68   |   60.16   |   74.04   |
> |        | PatchContrast    |     72.97    |   84.67   |   59.92   | **74.33** |
> |        | GPC (Ours)       |   **73.58**  |   84.68   | **62.06** |   73.99   |
>
> STRL: Huang  et al. "Spatio-temporal self-supervised representation learning for 3d point clouds." ICCV. 2021.
>
> ALSO: Boulch et al. "Also: Automotive lidar self-supervision by occupancy estimation." CVPR. 2023.
>
> GCC-3D: Liang et al. "Exploring geometry-aware contrast and clustering harmonization for self-supervised 3d object detection." ICCV. 2021.
>
> PatchContrast: Shrout et al., "PatchContrast: Self-Supervised Pre-training for 3D Object Detection." arXiv preprint arXiv:2308.06985.
>
> **Writing: decorative math.**
> We appreciate your feedback. We will try to squeeze Sect. 3 so we can include some rebuttal responses in the main paper.
>
> That said, we respectfully think the math on page 4 is meaningful. While the concepts of color variation (e.g., the same object can have different color options) and hints may sound intuitive in hindsight, we think a more formal description of the problem and the solution is needed for the soundness of our paper. In particular, Sect. 3.1 justifies why the color variation cannot be resolved by designing a more complex neural network or training objective and why adding hints is an effective solution. Our ablation studies in Sect. 4.3, in particular, Table 8 and Fig. 4, further validate the importance of adding hints for pre-training with colorization.

---

> ### Author Response · Authors · 2023-11-23
> **Rebuttal (Part 2)**
>
> **nuScenes dataset.**
> We thank the reviewer for the suggestion. We have provided some preliminary results of segmentation in our first response, *Downstream tasks.* Meanwhile, we have tried our best to conduct 3D object detection experiments of pre-training on NuScenes and fine-tuning on 5% NuScenes, using CenterPoint. The results are summarized in the table below. The performance of GPC is close to the SOTA, TriCC, and better than previous algorithms. Please note that the performance shown here could be suboptimal due to the time constraint of the rebuttal period.
>
> | Method        |    mAP   |    NDS   |
> |---------------|:--------:|:--------:|
> | scratch       |   38.0   |   44.3   |
> | PointContrast |   39.8   |   45.1   |
> | GCC-3D        |   41.1   |   46.8   |
> | SLidR         |   43.3   |   52.4   |
> | TriCC         | **44.6** | **54.4** |
> | GPC (Ours)    |   44.1   |   53.3   |
>
> TriCC: Pang et al. "Unsupervised 3D Point Cloud Representation Learning by Triangle Constrained Contrast for Autonomous Driving." CVPR. 2023.
>
> SLidR: Sautier et al. "Image-to-lidar self-supervised distillation for autonomous driving data." CVPR. 2022.
>
>
> ***In light of our clarification and additional results, we would like to ask if the reviewer will reconsider the rating. Thank you.***

---

> > ### Comment · Reviewer_anv1 · 2023-11-23
> >
> > Thanks for carrying out the high-quality rebuttal, and the reviewer appreciates the efforts. One more question: are there any downstream results on Waymo Open Dataset? or the results on nuScenes with more labeled data for fine-tuning? A general issue for pre-trained methods is that the improvement is very minor given large-scale labeled data.

---

> > > ### Author Response · Authors · 2023-11-23
> > > **Rebuttal (Part 3)**
> > >
> > > Thank you for the additional comments and questions. Unfortunately, with running all of our computational resources 24/7, this is the best we can provide at this moment. However, we will indeed provide more results in the final version.
> > >
> > > We agree that, in general, the effect of pre-training will diminish when labeled data available for fine-tuning are sufficient. However, this is exactly why GPC excels: when all methods yield similar results of fine-tuning on the entire dataset, GPC is particularly strong when we don’t have such a large amount of labeled data. We have validated this with the results shown in the main paper, as well as the new presented here. GPC consistently outperforms training from scratch with the entire dataset when fine-tuning on just 20% of the labeled data. We humbly think this is one of the most important advantages of pre-training, especially since acquiring 3D labels is expensive.

---

> > > > ### Comment · Reviewer_anv1 · 2023-11-23
> > > >
> > > > Thanks, I understand the situation. I believe this paper is beyond the acceptance threshold. The authors should enhance the results on nuScenes and Waymo in the revised version.

---

### Official Review · Reviewer_cL42 · 2023-10-31

**Soundness:** 2 fair
**Presentation:** 3 good
**Contribution:** 3 good
**Rating:** 6
**Confidence:** 4

**Summary:**

This paper focuses on an interesting topic - pretraining LiDAR networks by learning colorization.  To achieve LiDAR point colorization, this work proposes Grounded Point Colorization (GPC), which incorporates color as "context" by using GT colors during colorization.  The reported experiments seem to show it is effective to use colorization for LiDAR network pretraining.

**Strengths:**

1. It is a new research idea - using colorization as a pre-train strategy for LiDAR networks.

2. It is interesting to see that colorization might be effective for LiDAR-based network pretraining.

**Weaknesses:**

1. The authors use point-level, especially for the ground category, for pretraining, but lack validation of its effectiveness in instance-level 3D object detection tasks.

2. They employ a multimodal pretraining approach (camera + LiDAR), while the compared methods are all unimodal (LiDAR-only).

3. They did not conduct linear probing, so it's challenging to determine whether pretraining itself significantly improves performance or if enhancements were added during fine-tuning.

4. Many details related to the proposed method and experiments have not been provided, which could confuse the readers.
I have a few other questions listed in the next section.

**Questions:**

1. The camera and point cloud perspectives are inconsistent, which means that only a portion of points can find corresponding color information. It's somewhat surprising that this pretraining can significantly boost fine-tuning performance, especially in the case of KITTI, which only has a front-facing camera. Maybe the authors could provide a detailed analysis of why using colorization for pretraining is so effective.

2. The authors claim that GPC + 20% fine-tuning can surpass 100% of training from scratch. However, it seems they haven't mentioned how they selected this 20%. If it's a random 20%, then the current LaserMix approach can achieve nearly the same level of fully supervised performance.

---

> ### Author Response · Authors · 2023-11-23
> **Rebuttal**
>
> **Lack of validation in instance-level 3D object detection tasks.**
> We respectfully think there might be a misunderstanding. While we pre-train with point-level colorization, our main experimental results (cf. Table 1–9) are about instance-level 3D object detection. We will make sure we clarify this in the final version. In short, during pre-training, we leverage colors and the hint mechanism to equip the model backbone with the semantic cues that facilitate the subsequent fine-tuning for 3D object detection.
>
> **Multimodal pretraining approach.**
> Thank you for the comment. In our original submission, we indeed compared with other multimodal pre-training in Table 3. Please also see the paragraph *Learning with Images* on page 8 of the main paper.
>
> Here, we further provide the results of fine-tuning on 100% KITTI, using the PV-RCNN model.
> All the compared methods in the following table use multimodal pre-training, including the current SOTA: TriCC. GPC performs the best overall.
>
> |            |   easy   |   mod.   |   hard   |
> |------------|:--------:|:--------:|:--------:|
> | SLidR      |   82.9   |   71.9   |   68.0   |
> | TriCC      |   84.1   |   73.3   |   69.4   |
> | GPC (Ours) | **84.4** | **73.6** | **69.7** |
>
> TriCC: Pang et al. "Unsupervised 3D Point Cloud Representation Learning by Triangle Constrained Contrast for Autonomous Driving." CVPR. 2023.
>
> SLidR: Sautier et al. "Image-to-lidar self-supervised distillation for autonomous driving data." CVPR. 2022.
>
> **Linear probing.**
> Thanks for the comment. We note that while linear probing is usually used to evaluate pre-training on classification tasks, it may not be suitable for the downstream task of 3D object detection. For 3D object detection, the detection head added on top of the backbone contains multiple neural network layers and is trained for multiple tasks (e.g., regression and classification). In this paper, we follow the convention in this branch of study (e.g., PatchContrast, ProposalContrast, etc.) to perform full fine-tuning to compare with existing SOTA algorithms.
>
> **Details about the proposed method.**
> Thank you for pointing out the problem. Due to the space limit, we keep some details in the supplementary. We will incorporate more details and discussion in the final version.
>
> **Inconsistency of camera and point cloud.**
> Thanks for the question. We provided a detailed discussion regarding this question in the first response (i.e., *color assignment*) to Reviewer Hnni.
>
> Specifically about the KITTI dataset, since the 3D object detection metric only considers objects that appear within the frontal camera's field of view (Ref. [1]), it is a convention to input only the corresponding frontal-view LiDAR points into the 3D object detector in both training and testing (Ref. [2]).
>
> Regarding why using colorization for pre-training is effective, we have provided several analyses (cf. Sect. 4.3, S2, and S4) and arguments (cf. Sect. 1 and 3) in the main paper and supplementary. We will be happy to conduct additional analyses in the final version. We note that pre-training with naive colorization is suboptimal (cf. Table 6 with the MSE loss and Table 8 without hints). Our key contribution is identifying the intrinsic difficulty (cf. Sect. 3.1 and 3.2) and proposing GPC as the appropriate way to use color for pre-training.
>
> [1] https://www.cvlibs.net/datasets/kitti/eval_object.php?obj_benchmark=3d
>
> [2] https://github.com/open-mmlab/OpenPCDet/blob/master/tools/cfgs/dataset_configs/kitti_dataset.yaml#L17
>
> **Comparison to LaserMix at 20% fine-tuning.**
> Thanks for providing the reference. We carefully read it (Ref. [3]) and found that the focus differs from ours. LaserMix focuses on LiDAR semantic segmentation, while we focus on 3D object detection. We could not find its performance of 3D object detection. The fine-tuning subsets used in our paper are randomly sampled. However, we provide the results of repeated experiments in Appendix S1, showing that the improvement is consistent regardless of which data are sampled for fine-tuning. Our claim about the 20% fine-tuning is just to explain the effectiveness of the proposed method and demonstrate the potential benefits of reducing labeling efforts. It is worth noting that, at the time of our paper submission, the SOTA algorithm, ProposalContrast, achieved only an overall moderate performance of 66.2% when fine-tuned on 20% labeled data (cf. Table 2). In contrast, GPC achieved 70.1%, better than 69.5% achieved by training from scratch using the entire dataset. The same trend is found in our new results using PV-RCNN as the 3D model (cf. the response *More comparisons in Table 1 and Table 2* to Reviewer Hnni): GPC performs fairly well when fine-tuning with 20% data.
>
> [3] Kong et al. "Lasermix for semi-supervised lidar semantic segmentation." CVPR. 2023.
>
> ***In light of our clarification and additional results, we would like to ask if the reviewer will reconsider the rating. Thank you.***

---

### Official Review · Reviewer_nb3N · 2023-11-01

**Soundness:** 3 good
**Presentation:** 3 good
**Contribution:** 3 good
**Rating:** 8
**Confidence:** 4

**Summary:**

The paper presents a colorization-based pretraining approach for 3D LiDAR detectors. The idea is to use aligned RGB images to extract ground truth colors for a LiDAR point cloud and to then pretrain a network with the task of point cloud colorization. The paper empirically shows that a naive direct approach to this would actually harm the downstream detection performance, given that it would force the network to memorize specific scenes as the color of points could for example depend on the weather, the time of day. Furthermore, the color of many objects is not well-defined by the shape, e.g. cars come in different colors. To overcome this issue, the paper proposes to provide 20% of the points with the ground truth colors, thus allowing the network to focus on learning how to propagate these colors across semantically connected segments in a scene. Furthermore, instead of directly regressing RGB colors, all the colors in a dataset are clustered into 128 clusters and a class-balanced classification loss is used to train the network. Empirically, using the proposed pretraining, the paper shows that several downstream detectors can get significant improvements, often requiring only a fraction of the annotations to match the fully-supervised approach.

**Strengths:**

- The paper presents a interesting and quite orthogonal approach to most other self-supervised pretraining methods for point cloud-based detection. Many design choices makes sense to me.
- Empirical results look quite strong.
- Important aspects are ablated and I very much like the fact that the paper first shows that naive colorization actually harms the performance.
- The paper was a fun and refreshing read.

**Weaknesses:**

- The main weakness that I see is with the two baseline approaches, why did you focus on Point R-CNN and IA-SSD? There seem to be significantly better performing methods out there. These might not be point-based, but the paper also shows that the method transfers to voxel-based methods. So I'm mainly wondering what happens when we look at current state-of-the-art methods? Can we still gain something from colorization-based pretraining for those methods? Or is there some explicit reason why stronger methods cannot be considered?
- Secondly, the related work is a bit lacking. While in general it covers the considered areas nicely, I'm missing related work on image colorization based pretraining, image colorization with seeds and general point cloud colorization. A quick google search reveals that all of these things exist. While I don't think they reveal major conflicts that should lead to the paper being rejected, I do think they are highly related and should be discussed. I'll name a few below, but obviously these papers are cited by many other papers and I'm sure for all three categories you can find some more relevant related work.
  - Colorful Image Colorization by Zhang et al. looked at image colorization as a pretext task.
  - Point2color: 3D Point Cloud Colorization Using a Conditional Generative Network and Differentiable Rendering for Airborne LiDAR by Shinohara et al. looks at point cloud colorization with a GAN setup. This could actually be a very interesting alternative to the seeding proposed here.
  - Automated Colorization of a Grayscale Image With Seed Points Propagation by Wan et al. looks at seed-based image colorization. While different, it could be interesting to discuss such differences.
- While you compare to ProposalContrast and DepthContrast, I find the discussion of the results quite lacking. Obviously they outperforms the proposed method in the case of cyclist and car, but only the strong performance of GPC for the pedestrian class pulls the overall score up. It would be good to at least talk about this a bit in the discussion of Table 2.
- The ablation regarding the different loss variants is not that great. Again the discussion seems to be based on the overall score, but per class there are quite some significant differences. In general I am not completely convinced by this ablation, given these are completely different losses with potentially vastly different magnitudes. An actual fair comparison would require a new tuning of the learning rate, or potentially a normalization of the loss values with some theoretical maximum? Did you do something in this direction, or did you just replace the loss and assume that everything else stays the same? Also the SL1 regression loss seems better than CE loss in quite some cases, what about a balanced SL1? Could that be better?

**Questions:**

- How did you sample the points that get a ground truth? Is this completely random? Or do you do some more informed sampling, ensuring that at least 1 sample is present for each cluster (if available in an image)? I guess this is not super crucial, but it could potentially reduce the amount of needed points and thus boost scores further since the network has to predict more.
- Did you consider joint training? While we often see a strict pretraining and finetuning setup, think this could easily be trained jointly, maybe also improving results?
- In the introduction you write "A labeled example of 3D object detection comprises a point cloud and a set of bounding boxes. While the format is typical, there is no explicit connection between the input data and labels." I'm rather confused by this. To me there is definitely a direct connection between the input data and the labels and I don't understand how this is meant.

In general, while I enjoyed reading the paper, some sentences are a bit broken due to small grammar mistakes or missing articles. Please run this through a grammar checker or get a native speaker to proofread it.

---

> ### Author Response · Authors · 2023-11-23
> **Rebuttal (Part 1)**
>
> **Baseline approaches.** In the main paper, we aimed to present our colorization idea in a clear and accessible way, allowing readers to grasp the core concept easily. Point-based detectors (e.g., Point R-CNN and IA-SSD) were chosen for the purpose. That said, our GPC can definitely be applied to voxel-based detectors, after a voxel-to-point operation. (Such an operation has been used in LiDAR point segmentation when a voxel-based backbone is adopted.) To illustrate this, we have conducted 3D object detection experiments with PV-RCNN on Waymo -> KITTI. The results are summarized in the table below. GPC consistently outperforms the existing algorithms. We appreciate your suggestion and we will incorporate these new results into the final version.
>
> | Labels |      Method      | Overall mod. |  Car mod. | Ped. mod. | Cyc. mod. |
> |:------:|:----------------:|:------------:|:---------:|:---------:|:---------:|
> |   20%  | scratch          |     66.71    |   82.52   |   53.33   |   64.28   |
> |        | ProposalContrast |     68.13    | **82.65** |   55.05   |   66.68   |
> |        | PatchContrast    |     70.75    |   82.63   |   57.77   | **71.84** |
> |        | GPC (Ours)       |   **71.37**  | **82.65** | **59.87** |   71.59   |
> |   50%  | scratch          |     69.63    |   82.68   |   57.10   |   69.12   |
> |        | ProposalContrast |     71.76    |   82.92   |   59.92   |   72.45   |
> |        | PatchContrast    |     72.39    |   84.47   |   60.76   |   71.94   |
> |        | GPC (Ours)       |   **72.75**  | **84.58** | **61.06** | **72.61** |
> |  100%  | scratch          |     70.57    |   84.50   |   57.06   |   70.14   |
> |        | GCC-3D           |     71.26    |     -     |     -     |     -     |
> |        | STRL             |     71.46    |   84.70   |   57.80   |   71.88   |
> |        | PointContrast    |     71.55    |   84.18   |   57.74   |   72.72   |
> |        | ProposalContrast |     72.92    | **84.72** |   60.36   |   73.69   |
> |        | ALSO             |     72.96    |   84.68   |   60.16   |   74.04   |
> |        | PatchContrast    |     72.97    |   84.67   |   59.92   | **74.33** |
> |        | GPC (Ours)       |   **73.58**  |   84.68   | **62.06** |   73.99   |
>
> STRL: Huang  et al. "Spatio-temporal self-supervised representation learning for 3d point clouds." ICCV. 2021.
>
> ALSO: Boulch et al. "Also: Automotive lidar self-supervision by occupancy estimation." CVPR. 2023.
>
> GCC-3D: Liang et al. "Exploring geometry-aware contrast and clustering harmonization for self-supervised 3d object detection." ICCV. 2021.
>
> PatchContrast: Shrout et al., "PatchContrast: Self-Supervised Pre-training for 3D Object Detection." arXiv preprint arXiv:2308.06985.

---

> ### Author Response · Authors · 2023-11-23
> **Rebuttal (Part 2)**
>
> **More related work about colorization.** We sincerely appreciate your valuable comments and references. We will certainly cite and discuss them, and include more papers on image colorization-based pre-training, image colorization with seeds, and general point cloud colorization. In the following, we provide a short summary and discussion of the literature.
>
> Image colorization has been an active research area in computer vision [1-13], predicting colorful versions of grayscale images. Typically, this task relies on human-related cues such as scribbles [9], exemplars [7], or texts [8], as well as external priors like pre-trained detectors [3], generative models [10], or superpixels [11, 18], to achieve realistic and diverse colorization. [14-17] have extended the task to 3D point clouds. Specifically, [13, 16] uses a GAN-style loss, learning a discriminator to differentiate real colorful images and colorized images and using it to train the colorization model. This provides an alternative to using hints to resolve the color variation problem.
>
> Instead of treating colorization as the ultimate task, [1, 12, 13] have explored colorization as a pre-training task for 2D detection and segmentation. We note that [1, 12] do not specifically address the color variation problem.
>
> In our paper, we introduce colorization as the pre-training task to equip the point cloud representation with semantic cues useful for the downstream 3D LiDAR-based detectors. We identify color variation as the intrinsic challenge and propose *inserting color hints into the output of the backbone* to ground the colorization taking place in the color decoder (cf. Figure 2). We note that such a design is crucial—we cannot add color hints as input to the backbone as the downstream task takes only the LiDAR points as input. In contrast, for colorization methods not for pre-training, the hints can be directly added to the input gray-level image [9]. [18] proposed a hybrid method to first predict the color of each superpixel pixel using a neural network, and propagated them to nearby pixels using a hand-crafted method (e.g., 2D-PCA). In contrast, our GPC learns to propagate colors in an end-to-end fashion by directly minimizing the colorization loss, which is the key to benefit downstream 3D detection.
>
> [1] Zhang, Richard, Phillip Isola, and Alexei A. Efros. "Colorful image colorization." ECCV, 2016.
>
> [2] Larsson, Gustav, Michael Maire, and Gregory Shakhnarovich. "Learning representations for automatic colorization." ECCV. 2016.
>
> [3] Su, Jheng-Wei, Hung-Kuo Chu, and Jia-Bin Huang. "Instance-aware image colorization." CPVR. 2020.
>
> [4] Kumar, Manoj, Dirk Weissenborn, and Nal Kalchbrenner. "Colorization transformer." arXiv. 2021.
>
> [5] Weng, Shuchen, et al. "CT 2: Colorization transformer via color tokens." ECCV. 2022.
>
> [6] Ji, Xiaozhong, et al. "ColorFormer: Image colorization via color memory assisted hybrid-attention transformer." ECCV. 2022.
>
> [7] He, Mingming, et al. "Deep exemplar-based colorization." TOG. 2018.
>
> [8] Chang, Zheng, et al. "L-CoIns: Language-Based Colorization With Instance Awareness." CVPR. 2023.
>
> [9] Levin, Anat, Dani Lischinski, and Yair Weiss. "Colorization using optimization." ACM SIGGRAPH 2004 Papers. 2004. 689-694.
>
> [10] Kim, Geonung, et al. "Bigcolor: colorization using a generative color prior for natural images." ECCV. 2022.
>
> [11] Xia, Menghan, et al. "Disentangled image colorization via global anchors." TOG. 2022.
>
> [12] Larsson, Gustav, Michael Maire, and Gregory Shakhnarovich. "Colorization as a proxy task for visual understanding." CVPR. 2017.
>
> [13] Treneska, Sandra, et al. "Gan-based image colorization for self-supervised visual feature learning." Sensors 22.4 (2022): 1599.
>
> [14] Cao, Xu, and Katashi Nagao. "Point cloud colorization based on densely annotated 3d shape dataset." MultiMedia Modeling: 25th International Conference. 2019.
>
> [15] Liu, Jitao, Songmin Dai, and Xiaoqiang Li. "Pccn: Point cloud colorization network." ICIP. 2019.
>
> [16] Shinohara, Takayuki, Haoyi Xiu, and Masashi Matsuoka. "Point2color: 3D point cloud colorization using a conditional generative network and differentiable rendering for airborne LiDAR." CVPR. 2021.
>
> [17] Gao, Rongrong, et al. "Scene-level Point Cloud Colorization with Semantics-and-geometry-aware Networks." ICRA. 2023.
>
> [18] Wan, Shaohua, et al. "Automated colorization of a grayscale image with seed points propagation." IEEE Transactions on Multimedia 22.7 (2020): 1756-1768.

---

> ### Author Response · Authors · 2023-11-23
> **Rebuttal (Part 3)**
>
> **Loss ablation for pre-training.** We apologize for the confusion. Indeed, different losses require different hyperparameter tuning. Actually, we conducted comprehensive grid searches for pre-training learning rates and fine-tuning learning rates (i.e. 0.0001 to 0.01), as well as other optimization-related hyperparameters. What we reported in Table 6 of the main paper is the *best performance* of each loss. We will clarify the ablation and provide more discussion in the final version. Thank you for bringing this to our attention.
>
> Regarding regression, it is known to be sensitive to outliers. In contrast to MSE, SL1 is more commonly used in many computer vision tasks such as colorization and regressing the bounding box locations or offsets. Your suggestion of a balanced SL1 loss sounds interesting. If we understand it correctly, you mean to balance the SL1 loss similarly to the BalancedSoftmax. Unfortunately, unlike classification where one can easily calculate the balance factors based on the number of samples per discrete class, calculating such a factor for continuous values is not trivial (e.g., some binning may be needed).
>
> Instead, we explore the use of BalancedL1 (Ref [1]), which attempts to reduce the effect of outliers. The results are shown in the table below. To provide a snippet of how we did the grid search in our main paper, we have reported the top-5 performance from the grid search of more than 30 pairs of learning rates on the experiments of fine-tuning on 5% KITTI.
>
> In our case, the color distribution highly depends on the point distribution. For example, the gray ground and the larger objects like cars contain the majority of points. They are *inliers* for regression. MSE learns more towards outliers (smaller objects like pedestrians and cyclists), leading to better performance on them after fine-tuning (c.f. Table 6 in the main paper). In contrast, if this effect is reduced by SL1 or BalancedL1, the performance of cars can get better. Naive cross-entropy is also known to be biased towards major training examples in class-imbalanced learning. In our paper, the *per frame* re-weighting in BalancedSoftmax can reduce the effect of imbalance point distribution, leading to the best fine-tuning results for all classes.
>
>
> | loss                                            | pre-traing lr | fine-tuning lr | Overall mod. | Car mod. | Ped. mod. | Cyc. mod. |
> |-------------------------------------------------|:-------------:|:--------------:|:------------:|:--------:|:---------:|:---------:|
> | BalancedL1                                      |     0.002     |     0.0008     |     60.4     |   62.2   |    55.1   |    63.8   |
> | BalancedL1                                      |     0.001     |     0.0008     |     60.6     |   62.1   |    56.3   |    63.3   |
> | BalancedL1                                      |     0.001     |     0.0004     |     60.9     |   62.8   |    56.0   |    63.9   |
> | BalancedL1                                      |     0.002     |     0.0004     |     60.7     |   63.1   |    55.4   |    63.7   |
> | BalancedL1                                      |     0.001     |     0.0006     |     60.9     |   62.9   |    56.0   |    63.8   |
> | BlancedSoftmax (c.f. Table 6 in the main paper) |     0.001     |     0.0004     |   **62.7**   | **65.1** |  **57.4** |  **65.7** |
>
> [1] Pang, Jiangmiao, et al. "Libra r-cnn: Towards balanced learning for object detection." CVPR. 2019.
>
>
> **Point Sampling for hints.** Thank you for the interesting suggestion. In the main paper, the hints provided in the pre-training are completely random. We agree that, during pre-training, it may not be necessary to provide as many hints for points located on the ground, and instead, we should focus more on foreground points. One potential solution is to sample fewer points from the major color classes. For example, the number of sampled points per class is set proportional to the square root of the total number of points per class. We have implemented the idea and reported the results of fine-tuning on 5% KITTI in the table below. We found no significant difference with more balanced sampling. However, we agree that smarter sampling is indeed interesting for future research and exploration.
>
> |                        | Overall mod. | Car mod. | Ped. mod. | Cyc. mod. |
> |------------------------|:------------:|:--------:|:---------:|:---------:|
> | random sampling        |     62.7     |   65.1   |    57.4   |    65.7   |
> | more balanced sampling |     62.5     |   64.9   |    56.5   |    66.2   |

---

> ### Author Response · Authors · 2023-11-23
> **Rebuttal (Part 4)**
>
> **Comparison to ProposalContrast and DepthContrast.** We will surely include more discussions. In general, we found GPC particularly useful for the label-scarce setting (e.g., 5% and 20% in Table 2). We see a similar trend in our newly added PV-RCNN results above. Under the label-scarce setting, GPC performs particularly well on pedestrians and cyclists, while the gain of the latter case diminishes when more labeled data are available.
> We hypothesize that GPC performs particularly well on pedestrians due to the intrinsic difficulty. Compared to cars and cyclists, whose shapes and layouts are (partially) rigid, pedestrians have larger variations not only in colors but also in poses. This makes propagating the seed colors to the remaining points challenging, encouraging the backbone to learn more knowledge, e.g., establishing stronger connections between points that would in turn benefit downstream fine-tuning. In contrast, cars contain some parts that the model does not need additional hints to predict their colors. For example, as shown in Fig. 5(b) of the main paper, the model can learn the red tail lights without giving it any hints. We will include the discussion in the final version.
>
> **Joint training.** Thank you for the interesting question. In this paper, we mainly follow the pre-training and fine-tuning convention to study pre-training for 3D object detection. Joint training, or more precisely, multi-task learning, is another way to leverage multiple objectives.
> In our experience, joint training typically could outperform training the downstream task from scratch, but may not outperform pre-training and then fine-tuning. This is because, in joint training, the model backbone has to perform well on both tasks simultaneously, which may distract from the main objective. That said, we will try to include joint training in the final version.
>
> **The introduction about data and labels.** We apologize for the confusion. We mainly want to convey the idea that the connection between the input point cloud and the bounding boxes can not be straightforwardly learned by a black-box model. One way to make their connection more explicit is through first segmenting objects and then reading out their poses. We will carefully modify this sentence in the final version.
>
>
> **Grammar mistakes.** Thank you very much for pointing out the issue. In the final version, we will carefully review the entire paper to enhance its readability.

---

### Official Review · Reviewer_Hnni · 2023-11-01

**Soundness:** 4 excellent
**Presentation:** 4 excellent
**Contribution:** 4 excellent
**Rating:** 8
**Confidence:** 5

**Summary:**

This paper proposes a very interesting idea for boosting the performance of LiDAR-based 3D object detection by teaching the model to colorize LiDAR point clouds. The initial idea suffers from an inherent color variance issue, so the authors further propose the "hints" concept to directly provide ground-truth colors to some of the points that the model initially has to predict. This is a simple but effective idea grounded in both theory and experimental results. The proposed idea, namely GPC, demonstrates great performance in downstream 3D object detection tasks compared to benchmarks and has the potential to reduce the need for human annotations.

**Strengths:**

1. The idea of introducing colorization to pre-train a LiDAR-based 3D object detection model sounds novel, and the experimental results also demonstrate its effectiveness on benchmark datasets.
2. The method of providing ground-truth labels to seed points to reduce the inherent color variance issue also seems straightforward but very effective.
3. I love the way the authors write this paper because it provides a clear motivational flow, explaining why they believe colorization can bridge the gap between point cloud and bounding box prediction. They also describe how their first attempt failed due to the color invariance issue and further propose their thinking process and solution.

**Weaknesses:**

Please check Questions section for details.

**Questions:**

Overall, this is a very good paper, but I still have several concerns and hope to get the authors' clarification.

1. How can we ensure that the "color" assigned to each point is accurate? Color only exists in 2D images, and as mentioned by the authors, the color of a LiDAR point is retrieved by projecting it to the 2D image coordinate. One concern is that for those points out of the field of view of the camera (e.g., multiple points with different colors can be projected to the same 2D pixel), the color can be inaccurately labeled.

2. It would be great to have an ablation study regarding the number of color bins used.

3. The reweighted loss idea in Eq 5 sounds reasonable, but one concern is that it may make the pretrained model specific to a particular dataset, reducing its transferability since each dataset can have a different number of classes and class distribution. It would also be interesting to see how the pretrained model performs when it is trained on one dataset but fine-tuned on another (this experiment is not required in the rebuttal; providing insights into this is sufficient.)

4. Please include more recent state-of-the-art works for comparison in Table 1 and Table 2. While I can gauge the effectiveness of the proposed idea through existing comparisons, a more extensive comparison with more recent works on the same benchmark is also necessary. The leaderboard of these two benchmarks has several entries showing better performance, so it would be great to discuss the advantages of the proposed method when compared to them.

---

> ### Author Response · Authors · 2023-11-23
> **Rebuttal (Part 1)**
>
> We appreciate your positive feedback and your insightful questions. To clarify them, we have conducted more experiments and ablations, as shown below.
>
> **Color assignment.** Thanks for the question. For LiDAR points that are outside the camera's field of view and cannot be projected onto the image, we do not calculate loss on them during pre-training (cf. Eq 1 and Eq 4). Specifically for the KITTI dataset, since the 3D object detection metric only considers objects that appear within the camera's field of view, it is common to input only the corresponding LiDAR points into the 3D object detector (Ref. [1]).
>
> For LiDAR points that are within the camera’s field of view, we would require a good calibration between the camera and LiDAR to ensure correct color assignments. To our knowledge, existing datasets do provide well-calibrated data (e.g., for studying sensor fusion). Even if one wants to collect pre-training data on their own, there is well-documented code to calibrate LiDAR and cameras with checkerboards.
>
> We thank the reviewer for pointing out another issue: some LiDAR points may be projected onto the same pixel location, for example, due to sensor errors or decimal precision. We examined the KITTI dataset and found that, on average, about 0.18% of points have this issue. Compared to the label error rate in common datasets (e.g., 5.83% in ImageNet, 5.85% in CIFAR-100) (Ref. [2]), it is relatively low. To address this issue, one can remove these confusing points in pre-training. As shown in the following tables (with fine-tuning on 5% KITTI), we see no notable difference between removing these points or not. However, it is worth keeping this issue in mind if applying GPC to other datasets or tasks.
>
> |                                              | easy | moderate | hard |
> |----------------------------------------------|------|----------|------|
> | w/o removal (c.f. Table 1 in the main paper) | 77.7 | 62.7     | 56.7 |
> | w/ removal                                   | 77.8 | 62.3     | 56.2 |
>
> [1] https://github.com/open-mmlab/OpenPCDet/blob/master/tools/cfgs/dataset_configs/kitti_dataset.yaml#L17
>
> [2] Northcutt, Curtis G., Anish Athalye, and Jonas Mueller. "Pervasive label errors in test sets destabilize machine learning benchmarks." arXiv preprint arXiv:2103.14749 (2021).
>
>
> **Ablation study regarding the number of color bins.** Thanks for the question. We conducted a study on fine-tuning with 5% KITTI (cf. top row of Table 1 in the main paper), using 32, 64, and 128 color bins in pre-training. The results are summarized in the following table. We found no significant difference in the downstream detection task with different color bins in pre-training. We hypothesize two reasons. First, our pre-training goal is to equip the backbone with the knowledge to identify which subset of points should possess the same color and be segmented together. In this sense, it could be less sensitive to the number of color bins as long as we have enough of them. (As in Fig. 3, images with 32 bins still show high contrast between objects and the background.) Second, the pre-trained backbone will still be optimized with the downstream task data. We will include the ablation in the final version.
>
> | bin | easy | mod. | hard |
> |-----|------|------|------|
> | 32  | 76.0 | 62.6 | 56.8 |
> | 64  | 76.8 | 62.2 | 56.1 |
> | 128 | 77.7 | 62.7 | 56.7 |
>
>
> **The reweighted loss vs. transferability.** This is an interesting question. We indeed have results with pre-training on Waymo and fine-tuning on KITTI (cf. Table. 2 in the main paper). GPC consistently outperforms baselines and performs similarly to pre-training with KITTI, demonstrating the transferability. We will provide more results with PV-RCNN in the next response.
>
> We thank you for the insights about transferability and the distribution shifts between datasets. (This issue also exists in 2D recognition, e.g., ImageNet pre-training transferred to COCO detection.) We respectfully think the reweighting loss could mitigate the distribution shift to facilitate transferability. As discussed in the literature on class-imbalanced learning and long-tailed recognition, learning a model without re-weighting would result in a model attempting to predict the major classes in the dataset—essentially capturing the dataset-specific bias. Learning with the re-weighting loss (cf. Eq 5) would reduce this effect. When a huge distribution shift exists between datasets, one can use the quantization bins obtained from the target dataset for pre-training with the source data.

---

> ### Author Response · Authors · 2023-11-23
> **Rebuttal (Part 2)**
>
> **More comparisons in Table 1 and Table 2.**
> We provide more experimental results of pre-training on Waymo and fine-tuning on KITTI (cf. Table 2), using PV-RCNN as the 3D detector. (We note that many SOTA results use PV-RCNN.) As shown in the following table, GPC consistently outperforms existing algorithms. The existing SOTA algorithms primarily utilize contrastive learning (e.g., ProposalContrast and PatchContrast). They require specific designs (i.e., proposal encoding module, patch extraction module, etc.) to identify meaningful regions to contrast. GPC, in contrast, offers a fresh perspective on pre-training for 3D detectors. It is straightforward in its approach yet remarkably effective. GPC is particularly advantageous when the labeled data are scarce. Regardless of the 3D model (i.e., PointRCNN or PV-RCNN) and the pre-training dataset (i.e., Waymo or KITTI), GPC’s performance of fine-tuning on just 20% of labeled data is consistently better than training from scratch with the entire dataset (cf. Table 1 and Table 2 and the following table). This advantage holds the potential to reduce the annotation effort required. We will include the new results and more discussion in the final version.
>
> | Labels |      Method      | Overall mod. |  Car mod. | Ped. mod. | Cyc. mod. |
> |:------:|:----------------:|:------------:|:---------:|:---------:|:---------:|
> |   20%  | scratch          |     66.71    |   82.52   |   53.33   |   64.28   |
> |        | ProposalContrast |     68.13    | **82.65** |   55.05   |   66.68   |
> |        | PatchContrast    |     70.75    |   82.63   |   57.77   | **71.84** |
> |        | GPC (Ours)       |   **71.37**  | **82.65** | **59.87** |   71.59   |
> |   50%  | scratch          |     69.63    |   82.68   |   57.10   |   69.12   |
> |        | ProposalContrast |     71.76    |   82.92   |   59.92   |   72.45   |
> |        | PatchContrast    |     72.39    |   84.47   |   60.76   |   71.94   |
> |        | GPC (Ours)       |   **72.75**  | **84.58** | **61.06** | **72.61** |
> |  100%  | scratch          |     70.57    |   84.50   |   57.06   |   70.14   |
> |        | GCC-3D           |     71.26    |     -     |     -     |     -     |
> |        | STRL             |     71.46    |   84.70   |   57.80   |   71.88   |
> |        | PointContrast    |     71.55    |   84.18   |   57.74   |   72.72   |
> |        | ProposalContrast |     72.92    | **84.72** |   60.36   |   73.69   |
> |        | ALSO             |     72.96    |   84.68   |   60.16   |   74.04   |
> |        | PatchContrast    |     72.97    |   84.67   |   59.92   | **74.33** |
> |        | GPC (Ours)       |   **73.58**  |   84.68   | **62.06** |   73.99   |
>
> STRL: Huang  et al. "Spatio-temporal self-supervised representation learning for 3d point clouds." ICCV. 2021.
>
> ALSO: Boulch et al. "Also: Automotive lidar self-supervision by occupancy estimation." CVPR. 2023.
>
> GCC-3D: Liang et al. "Exploring geometry-aware contrast and clustering harmonization for self-supervised 3d object detection." ICCV. 2021.
>
> PatchContrast: Shrout, Oren, et al. "PatchContrast: Self-Supervised Pre-training for 3D Object Detection." arXiv preprint arXiv:2308.06985. 2023.

---

> > ### Comment · Reviewer_Hnni · 2023-11-23
> > **Final review**
> >
> > I am writing to acknowledge that I have reviewed the authors' response to my concerns, and I believe they have adequately addressed all of them. I have also noticed that all reviewers shared the same concern regarding the lack of SOTA detectors for comparisons. However, with the newly added experimental results using PV-RCNN, I think that this concern has been addressed.
> >
> > I would like to maintain my original rating of 8 (accept, good paper) and thank authors for their efforts in their rebuttal.

---

### Author Response · Authors · 2023-11-23
**Rebuttal (General)**

We thank all reviewers for their valuable insights and suggestions. We are glad to see reviewers are positive about the paper. We have tried our best to conduct more experiments and ablations to address the questions. Some of the questions are indeed very interesting and could potentially lead to future research and exploration. We will include the new results and discussion in the final version. Thank you all!

---

> ### Comment · Reviewer_nb3N · 2023-11-23
>
> Thank you for all the additional comments. I'm voting for this paper to be accepted and hope the AC agrees! Looking forward to more papers like this in the future :)

---

### Meta-Review · Area_Chair_Tf91 · 2023-12-05

**Metareview:**

The paper proposes a pre-training framework for LIDAR-based detection. The framework uses a masked colorization scheme (referred to color hints by the authors). The proposed pre-training pipeline improves the performance on KITTI and Waymo datasets, especially when only a fraction of the training labels are available.
Summary of the strength and weaknesses (as noted by reviewers)
+ Novel twist on colorization for 3D detection
+ Well written
- Initial narrow evaluation (many of the more interesting Waymo and nuScenes results only came it late in the rebuttal)
- Impact more pronounced on the low-label regime

Overall, all reviewers recommend acceptance. The AC agrees.

**Justification For Why Not Higher Score:**

The most exciting results in the paper stem from evaluations on subsets of the labeled data (1% or 10%) or are evaluated in KITTI (which should have been retired years ago). This makes it hard to see a potential impact of the paper. There are some early nuScenes results in the rebuttal, but they came in too late with too little context to sway my assessment significantly.

**Justification For Why Not Lower Score:**

The idea is nice and nobody recommends rejection.

---

### Decision · Program_Chairs · 2024-01-16

Accept (poster)